# ACCELERATING SGD WITH MOMENTUM FOR OVER-PARAMETERIZED LEARNING

**Chaoyue Liu**
Department of Computer Science
The Ohio State University
Columbus, OH 43210
liu.2656@osu.edu

**Mikhail Belkin**
Department of Computer Science
The Ohio State University
Columbus, OH 43210
mbelkin@cse.ohio-state.edu

## ABSTRACT

Nesterov SGD is widely used for training modern neural networks and other machine learning models. Yet, its advantages over SGD have not been theoretically clarified. Indeed, as we show in this paper, both theoretically and empirically, Nesterov SGD with any parameter selection does *not* in general provide acceleration over ordinary SGD. Furthermore, Nesterov SGD may diverge for step sizes that ensure convergence of ordinary SGD. This is in contrast to the classical results in the deterministic setting, where the same step size ensures accelerated convergence of the Nesterov's method over optimal gradient descent.

To address the non-acceleration issue, we introduce a compensation term to Nesterov SGD. The resulting algorithm, which we call *MaSS*, converges for same step sizes as SGD. We prove that MaSS obtains an accelerated convergence rates over SGD for any mini-batch size in the linear setting. For full batch, the convergence rate of MaSS matches the well-known accelerated rate of the Nesterov's method.

We also analyze the practically important question of the dependence of the convergence rate and optimal hyper-parameters on the mini-batch size, demonstrating three distinct regimes: linear scaling, diminishing returns and saturation.

Experimental evaluation of MaSS for several standard architectures of deep networks, including ResNet and convolutional networks, shows improved performance over SGD, SGD+Nesterov and Adam.

## 1 INTRODUCTION

Many modern neural networks and other machine learning models are *over-parametrized* (5). These models are typically trained to have near zero training loss, known as *interpolation* and often have strong generalization performance, as indicated by a range of empirical evidence including (23; 3). Due to a key property of interpolation – *automatic variance reduction* (discussed in Section 2.1), stochastic gradient descent (SGD) with constant step size is shown to converge to the optimum of a convex loss function for a wide range of step sizes (12). Moreover, the optimal choice of step size $\eta^*$ for SGD in that setting can be derived analytically.

The goal of this paper is to take a step toward understanding *momentum-based* SGD in the interpolating setting. Among them, stochastic version of Nesterov's acceleration method (SGD+Nesterov) is arguably the most widely used to train modern machine learning models in practice. The popularity of SGD+Nesterov is tied to the well-known acceleration of the deterministic Nesterov's method over gradient descent (15). Yet, has not not theoretically clear whether Nesterov SGD accelerates over SGD.

As we show in this work, both theoretically and empirically, Nesterov SGD with any parameter selection does *not* in general provide acceleration over ordinary SGD. Furthermore, Nesterov SGD may diverge, even in the linear setting, for step sizes that guarantee convergence of ordinary SGD. Intuitively, the lack of acceleration stems from the fact that, to ensure convergence, the step size of SGD+Nesterov has to be much smaller than the optimal step size for SGD. This is in contrast to the deterministic Nesterov method, which accelerates using the same step size as optimal gradient descent. As we prove rigorously in this paper, the slow-down of convergence caused by the small step

size negates the benefit brought by the momentum term. We note that a similar lack of acceleration for the stochastic Heavy Ball method was analyzed in (9).

To address the non-acceleration of SGD+Nesterov, we introduce an additional compensation term to allow convergence for the same range of step sizes as SGD. The resulting algorithm, *MaSS* (Momentum-added Stochastic Solver)[1] updates the weights $\mathbf{w}$ and $\mathbf{u}$ using the following rules (with the compensation term underlined):

$$
\begin{aligned}
\mathbf{w}_{t+1} &\leftarrow \mathbf{u}_t - \eta_1 \tilde{\nabla} f(\mathbf{u}_t), \\
\mathbf{u}_{t+1} &\leftarrow (1+\gamma)\mathbf{w}_{t+1} - \gamma \mathbf{w}_t + \underline{\eta_2 \tilde{\nabla} f(\mathbf{u}_t)}.
\end{aligned}
\tag{1}
$$

Here, $\tilde{\nabla}$ represents the stochastic gradient. The step size $\eta_1$, the momentum parameter $\gamma \in (0, 1)$ and the compensation parameter $\eta_2$ are independent of $t$.

We proceed to analyze theoretical convergence properties of MaSS in the interpolated regime. Specifically, we show that in the linear setting MaSS converges exponen-

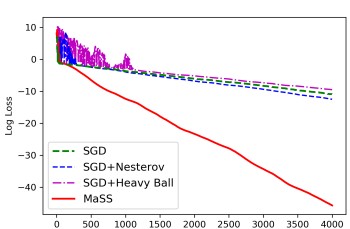

Figure 1: Non-acceleration of Nesterov SGD and fast convergence of MaSS.

tially for the same range of step sizes as plain SGD, and the optimal choice of step size for MaSS is exactly $\eta^*$ which is optimal for SGD. Our key theoretical result shows that MaSS has accelerated convergence rate over SGD. Furthermore, in the full batch (deterministic) scenario, our analysis selects $\eta_2 = 0$, thus reducing MaSS to the classical Nesterov's method (15). In this case our convergence rate also matches the well-known convergence rate for the Nesterov's method (15; 4). This acceleration is illustrated in Figure 1. Note that SGD+Nesterov (as well as Stochastic Heavy Ball) does not converge faster than SGD, in line with our theoretical analysis. We also prove exponential convergence of MaSS in more general convex setting under additional conditions.

We further analyze the dependence of the convergence rate $e^{-s(m)t}$ and optimal hyper-parameters on the mini-batch size $m$. We identify three distinct regimes of dependence defined by two critical values $m_1^*$ and $m_2^*$: linear scaling, diminishing returns and saturation, as illustrated in Figure 2. The convergence speed per iteration $s(m)$, as well as the optimal hyper-parameters, increase linearly as $m$ in the linear scaling regime, sub-linearly in the diminishing returns regime, and can only increase by a small constant factor in the saturation regime. The critical values $m_1^*$ and $m_2^*$ are derived analytically. We note that the intermediate "dimin-

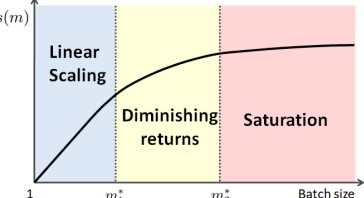

Figure 2: Convergence speed per iteration $s(m)$. Larger $s(m)$ indicates faster convergence per iteration.

ishing terurns" regime is new and is not found in SGD (12). To the best of our knowledge, this is the first analysis of mini-batch dependence for accelerated stochastic gradient methods.

We also experimentally evaluate MaSS on deep neural networks, which are non-convex. We show that MaSS outperforms SGD, SGD+Nesterov and Adam (10) both in optimization and generalization, on different architectures of deep neural networks including convolutional networks and ResNet (7).

The paper is organized as follows: In section 2, we introduce notations and preliminary results. In section 3, we discuss the non-acceleration of SGD+Nesterov. In section 4 we introduce MaSS and analyze its convergence and optimal hyper-parameter selection. In section 5, we analyze the mini-batch MaSS. In Section 6, we show experimental results.

## 1.1 RELATED WORK

Over-parameterized models have drawn increasing attention in the literature as many modern machine learning models, especially neural networks, are over-parameterized (5) and show strong generalization performance (16; 23; 2). Over-parameterized models usually result in nearly perfect fit (or interpolation) of the training data (23; 18; 3). Exponential convergence of SGD with constant step size under interpolation and its dependence on the batch size is analyzed in (12).

---

[1]Code url: https://github.com/ts66395/MaSS

There are a few works that show or indicate the non-acceleration of existing stochastic momentum methods. First of all, the work (9) theoretically proves non-acceleration of stochastic Heavy Ball method (SGD+HB) over SGD on certain synthetic data. Furthermore, these authors provide experimental evidence that SGD+Nesterov also converges at the same rate as SGD on the same data. The work (22) theoretically shows that, for sufficiently small step-sizes, SGD+Nesterov and SGD+HB is equivalent to SGD with a larger step size. However, the results in (22) do not exclude the possibility that acceleration is possible when the step size is larger. The work (11) concludes that "momentum hurts the convergence within the neighborhood of global optima", based on a theoretical analysis of SGD+HB. These results are consistent with our analysis of the standard SGD+Nesterov. However, this conclusion does not apply to all momentum methods. Indeed, we will show that MaSS provably improves convergence over SGD.

There is a large body of work, both practical and theoretical, on SGD with momentum, including (10; 8; 1). Adam (10), and its variant AMSGrad (17), are among the most practically used SGD methods with momentum. Unlike our method, Adam adaptively adjusts the step size according to a weight-decayed accumulation of gradient history. In (8) the authors proposed an accelerated SGD algorithm, which can be written in the form shown on the right hand side in Eq.8, but with different hyper-parameter selection. Their ASGD algorithm also has a tail-averaging step at the final stage. In the interpolated setting (no additive noise) their analysis yields a convergence rate of $O(Poly(\kappa, \tilde{\kappa}) \exp(-\frac{t}{9\sqrt{\kappa_1 \tilde{\kappa}}}))$ compared to $O(\exp(-\frac{t}{\sqrt{\kappa_1 \tilde{\kappa}}}))$ for our algorithm with batch size 1. We provide some experimental comparisons between their ASGD algorithm and MaSS in Fig. 4.

The work (21) proposes and analyzes another first-order momentum algorithm and derives convergence rates under a different set of conditions – the strong growth condition for the loss function in addition to convexity. As shown in Appendix F.3, on the example of a Gaussian distributed data, the rates obtained in (21) can be slower than those for SGD. In contrast, our algorithm is guaranteed to never have a slower convergence rate than SGD. Furthermore, in the same Gaussian setting MaSS matches the optimal accelerated full-gradient Nesterov rate.

Additionally, in our work we consider the practically important dependence of the convergence rate and optimal parameter selection on the mini-batch size, which to the best of our knowledge, has not been analyzed for momentum methods.

## 2 NOTATIONS AND PRELIMINARIES

Given dataset $\mathcal{D} = \{(\mathbf{x}_i, y_i)\}_{i=1}^n \subset \mathbb{R}^d \times \mathcal{C}$, we consider an objective function of the form $f(\mathbf{w}) = \frac{1}{n} \sum_{i=1}^n f_i(\mathbf{w})$, where $f_i$ only depends on a single data point $(\mathbf{x}_i, y_i)$. Let $\nabla f$ denote the exact gradient, and $\tilde{\nabla}_m f$ denote the unbiased stochastic gradient evaluated based on a mini-batch of size $m$. For simplicity, we also denote $\tilde{\nabla} f(\mathbf{w}) := \tilde{\nabla}_1 f(\mathbf{w})$. We use the concepts of strong convexity and smoothness of functions, see definitions in Appendix B.1. For loss function with $\mu$-strong convexity and $L$-smoothness, the condition number $\kappa$ is defined as $\kappa = L/\mu$.

In the case of the square loss, $f_i(\mathbf{w}) = \frac{1}{2}(\mathbf{w}^T \mathbf{x}_i - y_i)^2$, and the Hessian matrix is $H := \frac{1}{n} \sum_{i=1}^n \mathbf{x}_i \mathbf{x}_i^T$. Let $L$ and $\mu$ be the largest and the smallest non-zero eigenvalues of the Hessian respectively. Then the condition number is then $\kappa = L/\mu$ (note that zero eigenvalues can be ignored in our setting, see Section 4).

**Stochastic Condition Numbers.** For a quadratic loss function, let $\tilde{H}_m := \frac{1}{m} \sum_{i=1}^m \tilde{\mathbf{x}}_i \tilde{\mathbf{x}}_i^T$ denotes a mini-batch estimate of $H$. Define $L_1$ be the smallest positive number such that $\mathbb{E}\left[\|\tilde{\mathbf{x}}\|^2 \tilde{\mathbf{x}} \tilde{\mathbf{x}}^T\right] \preceq L_1 H$, and denote

$$L_m := L_1/m + (m-1)L/m. \tag{2}$$

Given a mini-batch size $m$, we define the *m-stochastic* condition number as $\kappa_m := L_m/\mu$.

Following (8), we introduce the quantity $\tilde{\kappa}$ (called statistical condition number in (8)), which is the smallest positive real number such that $\mathbb{E}\left[\|\tilde{\mathbf{x}}\|_{H^{-1}}^2 \tilde{\mathbf{x}} \tilde{\mathbf{x}}^T\right] \preceq \tilde{\kappa} H$.

**Remark 1.** We note that $L_1 \geq L$, since $\mathbb{E}\left[\|\tilde{\mathbf{x}}\|^2 \tilde{\mathbf{x}} \tilde{\mathbf{x}}^T\right] = \mathbb{E}[\tilde{H}_1^2] = H^2 + \mathbb{E}[(\tilde{H}_1 - H)^2] \succeq H^2$. Consequently, according to the definition of $L_m$ in Eq.2, we have

$$L_m \geq L, \ \forall m \geq 1; \quad L_m \to L, \ \text{as } m \to \infty. \tag{3}$$

Hence, the quadratic loss function is also $L_m$-smooth, for all $m \geq 1$. By the definition of $\kappa_m$, we also have

$$\kappa_m \geq \kappa, \; \forall m \geq 1; \quad \kappa_m \to \kappa, \text{ as } m \to \infty. \tag{4}$$

**Remark 2.** It is important to note that $\tilde{\kappa} \leq \kappa_1$, since $\mathbb{E}\left[\|\tilde{\mathbf{x}}\|_{H^{-1}}^2 \tilde{\mathbf{x}}\tilde{\mathbf{x}}^T\right] \preceq \frac{1}{\mu}\mathbb{E}\left[\|\tilde{\mathbf{x}}\|^2 \tilde{\mathbf{x}}\tilde{\mathbf{x}}^T\right] \preceq \kappa_1 H$.

## 2.1 CONVERGENCE OF SGD FOR OVER-PARAMETRIZED MODELS AND OPTIMAL STEP SIZE

We consider over-parametrized models that have zero training loss solutions on the training data (e.g., (23)). A solution $f_i(\mathbf{w})$ which fits the training data perfectly $f_i(\mathbf{w}) = 0$, $\forall i = 1, 2, \cdots, n$, is known as *interpolating*. In the linear setting, interpolation implies that the linear system $\{\mathbf{x}_i^T \mathbf{w} = y_i\}_{i=1}^n$ has at least one solution.

A key property of interpolation is *Automatic Variance Reduction* (AVR), where the variance of the stochastic gradient decreases to zero as the weight $\mathbf{w}$ approaches the optimal $\mathbf{w}^*$.

$$\text{Var}[\tilde{\nabla}_m f(\mathbf{w})] \preceq \|\mathbf{w} - \mathbf{w}^*\|^2 \mathbb{E}[(\tilde{H}_m - H)^2]. \tag{5}$$

For a detailed discussion of AVR see Appendix B.2.

Thanks to AVR, plain SGD with constant step size can be shown to converge exponentially for strongly convex loss functions (13; 19; 14; 12). The set of acceptable step sizes is $(0, 2/L_m)$, where $L_m$ is defined in Eq.2 and $m$ is the mini-batch size. Moreover, the optimal step size $\eta^*(m)$ of SGD that induces fastest convergence guarantee is proven to be $1/L_m$ (12).

## 3 NON-ACCELERATION OF SGD+NESTEROV

In this section we prove that SGD+Nesterov, with *any* constant hyper-parameter setting, does not generally improve convergence over optimal SGD. Specifically, we demonstrate a setting where SGD+Nesterov can be proved to have convergence rate of $(1 - O(1/\kappa))^t$, which is same (up to a constant factor) as SGD. In contrast, the classical accelerated rate for the deterministic Nesterov's method is $(1 - 1/\sqrt{\kappa})^t$.

We will consider the following two-dimensional data-generating **component decoupled model.** Fix an arbitrary $\mathbf{w}^* \in \mathbb{R}^2$ and randomly sample $z$ from $\mathcal{N}(0, 2)$. The data points $(\mathbf{x}, y)$ are constructed as follow:

$$\mathbf{x} = \begin{cases} \sigma_1 \cdot z \cdot \mathbf{e}_1, & \text{w.p. } 0.5, \\ \sigma_2 \cdot z \cdot \mathbf{e}_2, & \text{w.p. } 0.5, \end{cases} \quad \text{and } y = \langle \mathbf{w}^*, \mathbf{x} \rangle, \tag{6}$$

where $\mathbf{e}_1, \mathbf{e}_2 \in \mathbb{R}^2$ are canonical basis vectors, $\sigma_1 > \sigma_2 > 0$. It can be seen that $L_1 = \sigma_1^2$, $\kappa_1 = 6\sigma_1^2/\sigma_2^2$ and $\tilde{\kappa} = 6$ (See Appendix F.1). This model is similar to that used to analyze the stochastic Heavy Ball method in (9).

The following theorem gives a lower bound for the convergence of SGD+Nesterov, regarding the linear regression problem on the component decoupled data model. See Appendix C for the proof.

**Theorem 1** (Non-acceleration of SGD+Nesterov)**.** *Let $\{(\mathbf{x}_i, y_i)\}_{i=1}^n$ be a dataset generated according to the component decoupled model. Consider the optimization problem of minimizing quadratic function $\frac{1}{2n}\sum_i (\mathbf{x}_i^T \mathbf{w} - y_i)^2$. For any step size $\eta > 0$ and momentum parameter $\gamma \in (0, 1)$ of SGD+Nesterov with random initialization, with probability one, there exists a $T \in \mathbb{N}$ such that $\forall t > T$,*

$$\mathbb{E}\left[f(\mathbf{w}_t)\right] - f(\mathbf{w}^*) \geq \left(1 - \frac{C}{\kappa}\right)^t \left[f(\mathbf{w}_0) - f(\mathbf{w}^*)\right], \tag{7}$$

*where $C > 0$ is a constant.*

Compared with the convergence rate $(1 - 1/\kappa)^t$ of SGD (12), this theorem shows that SGD+Nesterov does not accelerate over SGD. This result is very different from that in the deterministic gradient scenario, where the classical Nesterov's method has a strictly faster convergence guarantee than gradient descent (15).

Intuitively, the key reason for the non-acceleration of SGD+Nesterov is a condition on the step size $\eta$ required for non-divergence of the algorithm. Specifically, when momentum parameter $\gamma$ is close to

1, $\eta$ is required to be less than $2(1-\gamma)/3 + O((1-\gamma)^2)$ (precise formulation is given in Lemma 1 in Appendix C). The slow-down resulting from the small step size necessary to satisfy that condition cannot be compensated by the benefit of the momentum term. In particular, the condition on the step-size of SGD+Nesterov excludes $\eta^*$ that achieves fastest convergence for SGD. We show in the following corollary that, with the step size $\eta^*$, SGD+Nesterov diverges. This is different from the deterministic scenario, where the Nesterov method accelerates using the same step size as gradient descent.

**Corollary 1.** *Consider the same optimization problem as in Theorem 1. Let step-size $\eta = \frac{1}{L_1} = \frac{1}{6\sigma_1^2}$ and acceleration parameter $\gamma \in [0.6, 1]$, then SGD+Nesterov, with random initialization, diverges with probability 1.*

We empirically verify the non-acceleration of SGD+Nesterov as well as Corollary 1, in Section 6 and Appendix F.2.

## 4 MaSS: Accelerating SGD

In this section, we propose MaSS, which introduces a compensation term (see Eq.1) onto SGD+Nesterov. We show that MaSS can converge exponentially for all the step sizes that result in convergence of SGD, i.e., $\eta \in (0, 2/L_m)$. Importantly, we derive a convergence rate $\exp(-t/\sqrt{\kappa_1\tilde{\kappa}})$, where $\tilde{\kappa} \leq \kappa_1$, for MaSS which is faster than the convergence rate for SGD $\exp(-t/\kappa_1)$. Moreover, we give an analytical expression for the optimal hyper-parameter setting.

For ease of analysis, we rewrite update rules of MaSS in Eq.1 in the following equivalent form (introducing an additional variable $\mathbf{v}$):

$$\begin{cases} \mathbf{w}_{t+1} \leftarrow \mathbf{u}_t - \eta_1 \tilde{\nabla} f(\mathbf{u}_t), \\ \mathbf{u}_{t+1} \leftarrow (1+\gamma)\mathbf{w}_{t+1} - \gamma\mathbf{w}_t + \eta_2 \tilde{\nabla} f(\mathbf{u}_t) \end{cases} \iff \begin{cases} \mathbf{w}_{t+1} \leftarrow \mathbf{u}_t - \eta \tilde{\nabla} f(\mathbf{u}_t), \\ \mathbf{v}_{t+1} \leftarrow (1-\alpha)\mathbf{v}_t + \alpha\mathbf{u}_t - \delta\tilde{\nabla} f(\mathbf{u}_t), \\ \mathbf{u}_{t+1} \leftarrow \frac{\alpha}{1+\alpha}\mathbf{v}_{t+1} + \frac{1}{1+\alpha}\mathbf{w}_{t+1}. \end{cases} \quad (8)$$

There is a bijection between the hyper-parameters $(\eta_1, \eta_2, \gamma)$ and $(\eta, \alpha, \delta)$, which is given by:

$$\gamma = (1-\alpha)/(1+\alpha), \quad \eta_1 = \eta, \quad \eta_2 = (\eta - \alpha\delta)/(1+\alpha). \quad (9)$$

**Remark 3** (SGD+Nesterov). In the literature, the Nesterov's method is sometimes written in a similar form as the R.H.S. of Eq.8. Since SGD+Nesterov has no compensation term, $\delta$ has to be fixed as $\eta/\alpha$, which is consistent with the parameter setting in (15).

**Assumptions.** We first assume square loss function, and later extend the analysis to general convex loss functions under additional conditions. For square loss function, the solution set $\mathcal{W}^* := \{\mathbf{w} \in \mathbb{R}^d | f(\mathbf{w}) = 0\}$ is an affine subspace in the parameter space $\mathbb{R}^d$. Given any $\mathbf{w}$, we denote its closest solution as $\mathbf{w}^* := \arg\min_{\mathbf{v} \in \mathcal{W}^*} \|\mathbf{w} - \mathbf{v}\|$, and define the error $\epsilon = \mathbf{w} - \mathbf{w}^*$. Be aware that different $\mathbf{w}$ may correspond to different $\mathbf{w}^*$, and that $\epsilon$ and (stochastic) gradients are always perpendicular to $\mathcal{W}^*$ (see discussion in Appendix B.3). Hence, no actual update happens along $\mathcal{W}^*$. For this reason, we can ignore zero eigenvalues of $H$ and restrict our analysis to the span of the eigenvectors of the Hessian with non-zero eigenvalues.

Based on the equivalent form of MaSS in Eq.8, the following theorem shows that, for square loss function in the interpolation setting, MaSS is guaranteed to have exponential convergence when hyper-parameters satisfy certain conditions.

**Theorem 2** (Convergence of MaSS). *Consider minimizing a quadratic loss function in the interpolation setting. Let $\mu$ be the smallest non-zero eigenvalue of the Hessian matrix $H$. Let $L_m$ be as defined in Eq.2. Denote $\tilde{\kappa}_m := \tilde{\kappa}/m + (m-1)/m$. In MaSS with mini batch of size $m$, if the positive hyper-parameters $\eta, \alpha, \delta$ satisfy the following two conditions:*

$$\alpha/\delta \leq \mu, \quad \alpha\delta\tilde{\kappa}_m + \eta(\eta L_m - 2) \leq 0, \quad (10)$$

*then, after $t$ iterations,*

$$\mathbb{E}\left[\|\mathbf{v}_t - \mathbf{w}^*\|_{H^{-1}}^2 + \frac{\delta}{\alpha}\|\mathbf{w}_t - \mathbf{w}^*\|^2\right] \leq (1-\alpha)^t \left(\|\mathbf{v}_0 - \mathbf{w}^*\|_{H^{-1}}^2 + \frac{\delta}{\alpha}\|\mathbf{w}_0 - \mathbf{w}^*\|^2\right).$$

*Consequently,*

$$\|\mathbf{w}_t - \mathbf{w}^*\|^2 \leq C \cdot (1-\alpha)^t,$$

*for some constant $C > 0$ which depends on the initialization.*

**Remark 4.** By condition Eq.10, the admissible step size $\eta$ is $(0, 2/L_m)$, exactly the same as SGD for interpolated setting (12).

**Remark 5.** One can easily check that the hyper-parameter setting of SGD+Nesterov does not satisfy the conditions in Eq.10.

*Proof sketch for Theorem 2.* Denote $\mathcal{F}_t := \mathbb{E}\left[\|\mathbf{v}_{t+1} - \mathbf{w}^*\|^2_{H^{-1}} + \frac{\delta}{\alpha}\|\mathbf{w}_{t+1} - \mathbf{w}^*\|^2\right]$, we show that, under the update rules of MaSS in Eq.8,

$$\mathcal{F}_{t+1} \leq (1-\alpha)\mathcal{F}_t + + \underbrace{(\alpha/\mu - \delta)}_{c_1}\|\mathbf{u}_t - \mathbf{w}^*\|^2 + \underbrace{\left(\delta^2\tilde{\kappa}_m + \delta\eta^2 L_m/\alpha - 2\eta\delta/\alpha\right)}_{c_2}\|\mathbf{u}_t - \mathbf{w}^*\|^2_H.$$

By the condition in Eq.10, $c_1 \leq 0, c_2 \leq 0$, then the last two terms are non-positive. Hence,

$$\mathcal{F}_{t+1} \leq (1-\alpha)\mathcal{F}_t \leq (1-\alpha)^{t+1}\mathcal{F}_0. \tag{11}$$

Using that $\|\mathbf{w}_t - \mathbf{w}^*\|^2 \leq \alpha\mathcal{F}_t/\delta$, we get the final conclusion. See detailed proof in Appendix D. □

**Hyper-parameter Selection.** From Theorem 2, we observe that the convergence rate is determined by $(1-\alpha)^t$. Therefore, larger $\alpha$ is preferred for faster convergence. Combining the conditions in Eq.10, we have

$$\alpha^2 \leq \eta(2 - \eta L_m)\mu/\tilde{\kappa}_m. \tag{12}$$

By setting $\eta^* = 1/L_m$, which maximizes the right hand side of the inequality, we obtain the optimal selection $\alpha^* = 1/\sqrt{\kappa_m\tilde{\kappa}_m}$. Note that this setting of $\eta^*$ and $\alpha^*$ determines a unique $\delta^* = \alpha^*/\mu$ by the conditions in Eq.10. In summary,

$$\eta^*(m) = \frac{1}{L_m}, \alpha^*(m) = \frac{1}{\sqrt{\kappa_m\tilde{\kappa}_m}}, \delta^*(m) = \frac{\eta^*}{\alpha^*\tilde{\kappa}_m}. \tag{13}$$

By Eq.9, the optimal selection of $(\eta_1, \eta_2, \gamma)$ would be:

$$\eta_1^*(m) = \frac{1}{L_m}, \quad \eta_2^*(m) = \frac{\eta_1^*\sqrt{\kappa_m\tilde{\kappa}_m}}{\sqrt{\kappa_m\tilde{\kappa}_m} + 1}\left(1 - \frac{1}{\tilde{\kappa}_m}\right), \quad \gamma^*(m) = \frac{\sqrt{\kappa_m\tilde{\kappa}_m} - 1}{\sqrt{\kappa_m\tilde{\kappa}_m} + 1}. \tag{14}$$

$\tilde{\kappa}_m$ is usually larger than 1, which implies that the coefficient $\eta_2^*$ of the compensation term is non-negative. The non-negative coefficient $\eta_2$ indicates that the weight $\mathbf{u}_t$ is "over-descended" in SGD+Nesterov and needs to be compensated along the gradient direction.

It is important to note that the optimal step size for MaSS as in Eq.13 is exactly the same as the optimal one for SGD (12). With such hyper-parameter selection given in Eq.14, we have the following theorem for optimal convergence:

**Theorem 3** (Acceleration of MaSS). *Under the same assumptions as in Theorem 2, if we set hyper-parameters in MaSS as in Eq.13, then after $t$ iteration of MaSS with mini batch of size $m$,*

$$\|\mathbf{w}_t - \mathbf{w}^*\|^2 \leq C \cdot (1 - 1/\sqrt{\kappa_m\tilde{\kappa}_m})^t, \tag{15}$$

*for some constant $C > 0$ which depends on the initialization.*

**Remark 6.** With the optimal hyper-parameters in Eq.13, the asymptotic convergence rate of MaSS is

$$O(e^{-t/\sqrt{\kappa_m\tilde{\kappa}_m}}), \tag{16}$$

which is faster than the rate $O(e^{-t/\kappa_m})$ of SGD (see (12)), since $\kappa_m \geq \tilde{\kappa}_m$.

**Remark 7** (MaSS Reduces to the Nesterov's method for full batch). In the limit of full batch $m \to \infty$, we have $\kappa_m \to \kappa$, $\tilde{\kappa}_m \to 1$, the optimal parameter selection in Eq.14 reduces to

$$\eta_1^* = 1/L, \quad \gamma^* = (\sqrt{\kappa} - 1)/(\sqrt{\kappa} + 1), \quad \eta_2^* = 0. \tag{17}$$

It is interesting to observe that, in the full batch (deterministic) scenario, the compensation term vanishes and $\eta_1^*$ and $\gamma^*$ are the same as those in Nesterov's method. Hence MaSS with the optimal hyper-parameter selection reduces to Nesterov's method in the limit of full batch. Moreover, the convergence rate in Theorem 3 reduces to $O(e^{-t/\sqrt{\kappa}})$, which is exactly the well-known convergence rate of Nesterov's method (15; 4).

**Extension to Convex Case.**  First, we extend the definition of $L_1$ to convex functions, $L_1 := \inf\{c \in \mathbb{R} \mid \mathbb{E}[\|\tilde{\nabla} f(\mathbf{w})\|^2] \leq 2c (f(\mathbf{w}) - f^*)\}$, and keep the definition of $L_m$ the same as Eq.2. It can be shown that these definitions of $L_m$ are consistent with those in the quadratic setting. We assume that the smallest positive eigenvalue of Hessian $H(\mathbf{x})$ is lower bounded by $\mu > 0$, for all $\mathbf{x}$.

**Theorem 4.** *Suppose there exists a $\frac{1}{L}$-strongly convex and $\frac{1}{\mu}$-smooth non-negative function $g :$ $\mathbb{R}^d \to \mathbb{R}$ such that $g(\mathbf{w}^*) = 0$ and $\langle \nabla g(\mathbf{x}), \nabla f(\mathbf{z}) \rangle \geq (1 - \epsilon)\langle \mathbf{x} - \mathbf{w}^*, \mathbf{z} - \mathbf{w}^* \rangle, \forall \mathbf{x}, \mathbf{z} \in \mathbb{R}^d$, for some $\epsilon > 0$. In MaSS, if the hyper-parameters are set to be:*

$$\eta = 1/(2L_m), \quad \alpha = (1 - \epsilon)/(2\kappa_m), \quad \delta = 1/(2L_m), \tag{18}$$

*then after $t$ iterations, there exists a constant $C$ such that, $\mathbb{E}[f(\mathbf{w}_t)] \leq C \cdot \left(1 - \frac{1 - \epsilon}{2\kappa_m}\right)^t$.*

## 5 LINEAR SCALING RULE AND THE DIMINISHING RETURNS

Based on our analysis, we discuss the effect of selection of mini-batch size $m$. We show that the domain of mini-batch size $m$ can be partitioned into three intervals by two critical points: $m_1^* = \min(L_1/L, \tilde{\kappa}), m_2^* = \max(L_1/L, \tilde{\kappa})$. The three intervals/regimes are depicted in Figure 2, and the detailed analysis is in Appendix G.

**Linear Scaling:** $m < m_1^*$**.** In this regime, we have $L_m \approx L_1/m, \kappa_m \approx \kappa_1/m$ and $\tilde{\kappa}_m \approx \tilde{\kappa}/m$. The optimal selection of hyper-parameters is approximated by:

$$\eta^*(m) \approx m \cdot \eta^*(1), \alpha^*(m) \approx m \cdot \alpha^*(1), \delta^*(m) \approx m \cdot \delta^*(1),$$

and the convergence rate in Eq.16 is approximately $O(e^{-m \cdot t/\sqrt{\kappa_1 \tilde{\kappa}}})$. This indicates linear gains in convergence when $m$ increases.

In the linear scaling regime, the hyper-parameter selections follow a *Linear Scaling Rule* (LSR): When the mini-batch size is multiplied by $k$, multiply *all* hyper-parameters $(\eta, \alpha, \delta)$ by $k$. This parallels the linear scaling rule for SGD which is an accepted practice for training neural networks (6).

Moreover, increasing $m$ results in linear gains in the convergence speed, i.e., one MaSS iteration with mini-batch size $m$ is almost as effective as $m$ MaSS iterations with mini-batch size 1.

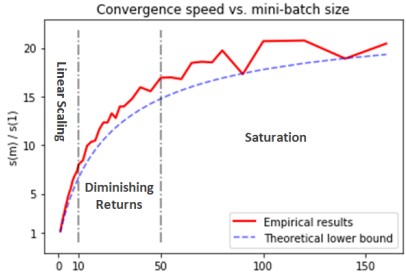

**Diminishing Returns:** $m \in [m_1^*, m_2^*)$  In this regime, increasing $m$ results in sublinear gains in convergence speed. One MaSS iteration with mini-batch size $m$ is less effective than $m$ MaSS iterations with mini-batch size 1.

Figure 3: Convergence speed per iteration $s(m)$. Larger $s(m)$ indicates faster convergence. Red solid curve: experimental results. Blue dash curve: theoretical lower bound $\sqrt{\kappa_m \tilde{\kappa}_m}$. Critical mini-batch sizes: $m_1^* \approx 10, m_2^* \approx 50$.

**Saturation:** $m \geq m_2^*$**.**  One MaSS iteration with mini-batch size $m$ is nearly as effective (up to a multiplicative factor of 2) as one iteration with full gradient.

This three regimes partition is different from that for SGD (12), where only linear scaling and saturation regimes present. An empirical verification of the dependence of the convergence speed on $m$ is shown in Figure 3. See the setup in Appendix G.

## 6 EMPIRICAL EVALUATION

**Synthetic Data.**  We empirically verify the non-acceleration of SGD+Nesterov and the fast convergence of MaSS on synthetic data. Specifically, we optimize the quadratic function $\frac{1}{2n} \sum_i (\mathbf{x}_i^T \mathbf{w} - y_i)^2$, where the dataset $\{(\mathbf{x}_i, y_i)\}_{i=1}^n$ is generated by the component decoupled model described in Section 3. We compare the convergence behavior of SGD+Nesterov with SGD, as well as our proposed method, MaSS, and several other methods: SGD+HB, ASGD (8). We select the best hyper-parameters from dense grid search for SGD+Nesterov (step-size and momentum parameter),

SGD+HB (step-size and momentum parameter) and SGD (step-size). For MaSS, we do not tune the hyper-parameters but use the hyper-parameter setting suggested by our theoretical analysis in Section 4; For ASGD, we use the setting provided by (8).

Fig. 4 shows the convergence behaviors of these algorithms on the setting of $\sigma_1^2 = 1, \sigma_2^2 = 1/2^9$. We observe that the fastest convergence of SGD+Nesterov is almost identical to that of SGD, indicating the non-acceleration of SGD+Nesterov. We also observe that our proposed method, MaSS, clearly outperforms the others. In Appendix F.2, we provide additional experiments on more settings of the component decoupled data, and Gaussian distributed data. We also show the divergence of SGD+Nesterov with the same step size as SGD and MaSS in Appendix F.2.

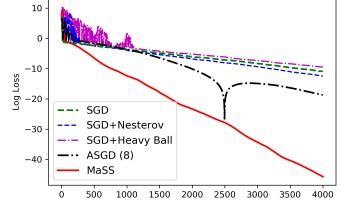

Figure 4: Comparison on component decoupled data. Hyperparameters: we use optimal parameters for SGD+Nesterov, SGD+HB and SGD; the setting in (8) for ASGD; Eq. 13 for MaSS.

**Real data: MNIST and CIFAR-10.** We compare the optimization performance of SGD, SGD+Nesterov and MaSS on the following tasks: classification of MNIST with a fully-connected network (FCN), classification of CIFAR-10 with a convolutional neural network (CNN) and Gaussian kernel regression on MNIST. See detailed description of the architectures in Appendix H.1. In all the tasks and for all the algorithms, we select the best hyper-parameter setting over dense grid search, except that we fix the momentum parameter $\gamma = 0.9$ for both SGD+Nesterov and MaSS, which is typically used in practice. All algorithms are implemented with mini batches of size $64$ for neural network training.

Fig. 5 shows the training curves of MaSS, SGD+Nesterov and SGD, which indicate the fast convergence of MaSS on real tasks, including the non-convex optimization problems on neural networks.

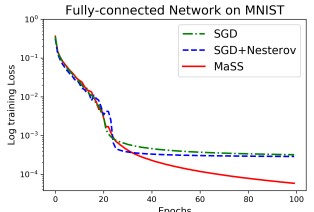 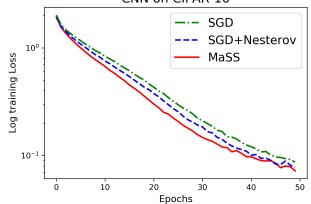 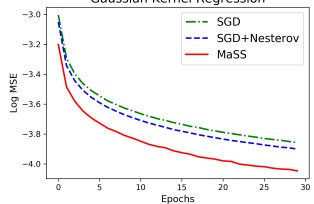

Figure 5: Comparison of SGD, SGD+Nesterov and MaSS on (left) fully-connected neural network, (middle) convolutional neural network, and (right) kernel regression.

**Test Performance.** We show that the solutions found by MaSS have good generalization performance. We evaluate the classification accuracy of MaSS, and compare with SGD, SGD+Nesterov and Adam, on different modern neural networks: CNN and ResNet (7). See description of the architectures in Appendix H.1. In the training processes, we follow the standard protocol of *data augmentation* and *reduction of learning rate*, which are typically used to achieve state-of-the-art results in neural networks. In each task, we use the same initial learning rate for MaSS, SGD and SGD+Nesterov, and run the same number of epochs (150 epochs for CNN and 300 epochs for ResNet-32). Detailed experimental settings are deferred to Appendix H.2.

Table 6 compares the classification accuracy of these algorithms on the test set of CIFAR-10 (average of 3 independent runs).

We observe that MaSS produces the best test performance. We also note that increasing initial learning rate may improves performance of MaSS and SGD, but degrades that of

|  | $\eta$ | SGD | SGD+Nesterov | MaSS | Adam† |
|---|---|---|---|---|---|
| CNN | 0.01 | 81.40% | 83.06% | **83.97%** | 82.65% |
|  | 0.3 | 82.41% | 75.79%♯ | **84.48%** |  |
| ResNet-32 | 0.1 | 91.92% | 92.60% | **92.77%** | 92.27% |
|  | 0.3 | 92.51% | 91.13%♯ | **92.71%** |  |

♯ Average of 3 runs that converge. Some runs diverge.
† Adam uses initial step size 0.001.

Table 1: Classification accuracy on test set of CIFAR-10.

SGD+Nesterov. Moreover, in our experiment, SGD+Nesterov with large step size $\eta = 0.3$ diverges in 5 out of 8 runs on CNN and 2 out of 5 runs on ResNet-32 (for random initialization), while MaSS and SGD converge on every run.

## ACKNOWLEDGEMENTS

This research was in part supported by NSF funding and a Google Faculty Research Award. GPUs donated by Nvidia were used for the experiments. We thank Ruoyu Sun for helpful comments concerning convergence rates. We thank Xiao Liu for helping with the empirical evaluation of our proposed method.

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

## A  PSEUDOCODE FOR MASS

---

**Algorithm 1** : *MaSS*–Momentum-added Stochastic Solver

---

   **Require**: Step-size $\eta_1$, secondary step-size $\eta_2$, acceleration parameter $\gamma \in (0, 1)$.
   **Initialize**: $\mathbf{u}_0 = \mathbf{w}_0$.
   **while**  not converged **do**
      $\mathbf{w}_{t+1} \leftarrow \mathbf{u}_t - \eta_1 \tilde{\nabla} f(\mathbf{u}_t)$,
      $\mathbf{u}_{t+1} \leftarrow (1 + \gamma)\mathbf{w}_{t+1} - \gamma\mathbf{w}_t - \eta_2 \tilde{\nabla} f(\mathbf{u}_t)$.
   **end while**
   **Output**: weight $\mathbf{w}_t$.

---

Note that the proposed algorithm initializes the variables $\mathbf{w}_0$ and $\mathbf{u}_0$ with the same vector, which could be randomly generated.

As discussed in section 4, MaSS can be equivalently implemented using the following update rules:

$$\mathbf{w}_{t+1} \quad \leftarrow \quad \mathbf{u}_t - \eta \tilde{\nabla} f(\mathbf{u}_t), \tag{19a}$$

$$\mathbf{v}_{t+1} \quad \leftarrow \quad (1 - \alpha)\mathbf{v}_t + \alpha\mathbf{u}_t - \delta \tilde{\nabla} f(\mathbf{u}_t), \tag{19b}$$

$$\mathbf{u}_{t+1} \quad \leftarrow \quad \frac{\alpha}{1 + \alpha}\mathbf{v}_{t+1} + \frac{1}{1 + \alpha}\mathbf{w}_{t+1}. \tag{19c}$$

In this case, variables $\mathbf{u}_0$, $\mathbf{v}_0$ and $\mathbf{w}_0$ should be initialized with the same vector.

There is a bijection between the hyper-parameters $(\eta_1, \eta_2, \gamma)$ and $(\eta, \alpha, \delta)$, which is given by:

$$\gamma = \frac{1 - \alpha}{1 + \alpha}, \quad \eta_1 = \eta, \quad \eta_2 = \frac{\eta - \alpha\delta}{1 + \alpha}. \tag{20}$$

## B  ADDITIONAL PRELIMINARIES

### B.1  STRONG CONVEXITY AND SMOOTHNESS OF FUNCTIONS

**Definition 1** (Strong Convexity). *A differentiable function $f : \mathbb{R}^d \to \mathbb{R}$ is $\mu$-strongly convex ($\mu > 0$), if*

$$f(\mathbf{x}) \geq f(\mathbf{z}) + \langle \nabla f(\mathbf{z}), \mathbf{x} - \mathbf{z} \rangle + \frac{\mu}{2}\|\mathbf{x} - \mathbf{z}\|^2, \quad \forall \mathbf{x}, \mathbf{z} \in \mathbb{R}^d. \tag{21}$$

**Definition 2** (Smoothness). *A differentiable function $f : \mathbb{R}^d \to \mathbb{R}$ is $L$-smooth ($L > 0$), if*

$$f(\mathbf{x}) \leq f(\mathbf{z}) + \langle \nabla f(\mathbf{z}), \mathbf{x} - \mathbf{z} \rangle + \frac{L}{2}\|\mathbf{x} - \mathbf{z}\|^2, \quad \forall \mathbf{x}, \mathbf{z} \in \mathbb{R}^d. \tag{22}$$

### B.2  AUTOMATIC VARIANCE REDUCTION

In the interpolation setting, one can write the square loss as

$$f(\mathbf{w}) = \frac{1}{2}(\mathbf{w} - \mathbf{w}^*)^T H(\mathbf{w} - \mathbf{w}^*) = \frac{1}{2}\|\mathbf{w} - \mathbf{w}^*\|_H^2. \tag{23}$$

A key property of interpolation is that the variance of the stochastic gradient of decreases to zero as the weight $\mathbf{w}$ approaches an optimal solution $\mathbf{w}^*$.

**Proposition 1** (Automatic Variance Reduction). For the square loss function $f$ in the interpolation setting, the stochastic gradient at an arbitrary point $\mathbf{w}$ can be written as

$$\tilde{\nabla}_m f(\mathbf{w}) = \tilde{H}_m(\mathbf{w} - \mathbf{w}^*) = \tilde{H}_m \boldsymbol{\epsilon}. \tag{24}$$

Moreover, the variance of the stochastic gradient

$$\text{Var}[\tilde{\nabla}_m f(\mathbf{w})] \preceq \|\boldsymbol{\epsilon}\|^2 \mathbb{E}[(\tilde{H}_m - H)^2]. \tag{25}$$

Since $\mathbb{E}[(\tilde{H}_m - H)^2]$ is independent of $\mathbf{w}$, the above proposition unveils a linear dependence of variance of stochastic gradient on the norm square of error $\boldsymbol{\epsilon}$. This observation underlies exponential convergence of SGD in certain convex settings (20; 13; 19; 14; 12).

### B.3 ON IGNORING ZERO EIGENVALUES OF THE HESSIAN

Consider the square loss function, $f(w) = \frac{1}{2n} \sum_i (\mathbf{x}_i^T \mathbf{w} - y_i)^2$. The (stochastic) gradient is

$$\tilde{\nabla}_m f(\mathbf{w}) = \frac{1}{m} \sum_{i=1}^m \mathbf{x}_i (\mathbf{x}_i^T \mathbf{w} - y_i) = \frac{1}{m} \sum_{i=1}^m \mathbf{x}_i \mathbf{x}_i^T (\mathbf{w} - \mathbf{w}^*) = H_m(\mathbf{w} - \mathbf{w}^*),$$

where the stochastic gradient is computed based on a randomly sampled batch of size $m$.

Recall that the solution set $\mathcal{W}^* := \{\mathbf{w} \in \mathbb{R}^d | f(\mathbf{w}) = 0\}$ is an affine subspace in the parameter space, and that $\mathbf{w}^*$ is the solution in $\mathcal{W}^*$ that is closest to $\mathbf{w}$. Hence, $\mathbf{w} - \mathbf{w}^*$ is perpendicular to $\mathcal{W}^*$, i.e., $\mathbf{w} - \mathbf{w}^* \in (\mathcal{W}^*)^\perp$. Note that $(\mathcal{W}^*)^\perp$ is an invariant space of $H_m$, hence $\tilde{\nabla}_m f(\mathbf{w}) = H_m(\mathbf{w} - \mathbf{w}^*) \in (\mathcal{W}^*)^\perp$. Hence the (stochastic) gradient is perpendicular to $\mathcal{W}^*$.

## C  PROOF OF THEOREM 1

The key proof technique is to consider the asymptotic behavior of SGD+Nesterov in the decoupled model of data when the condition number becomes large.

*Notations and proof setup.* Recall that the square loss function based on the component decoupled data $\mathcal{D}$, define in Eq.6, is in the interpolation regime, then for SGD+Nesterov, we have the recurrence relation

$$\mathbf{w}_{t+1} = (1 + \gamma)(1 - \eta\tilde{H})\mathbf{w}_t - \gamma(1 - \eta\tilde{H})\mathbf{w}_{t-1}, \tag{26}$$

where $\tilde{H} = \text{diag}((x^{[1]})^2, (x^{[2]})^2)$. It is important to note that each component of $\mathbf{w}_t$ evolves independently, due to the fact that $\tilde{H}$ is diagonal.

With $\epsilon := \mathbf{w} - \mathbf{w}^*$, we define for each component $j = 1, 2$ that

$$\Phi_{t+1}^{[j]} := \mathbb{E}\left[ \begin{pmatrix} \epsilon_{t+1}^{[j]} \\ \epsilon_t^{[j]} \end{pmatrix} \otimes \begin{pmatrix} \epsilon_{t+1}^{[j]} \\ \epsilon_t^{[j]} \end{pmatrix} \right], \quad j = 1, 2, \tag{27}$$

where $\epsilon^{[j]}$ is the $j$-th component of vector $\epsilon$.

The recurrence relation in Eq.26 can be rewritten as

$$\Phi_{t+1}^{[j]} = \mathcal{B}^{[j]} \Phi_t^{[j]}, \tag{28}$$

with

$$
\mathcal{B}^{[j]} =
$$
$$
\mathbb{E} \begin{pmatrix}
(1+\gamma)^2(1-\eta(\tilde{x}^{[j]})^2)^2 & -\gamma(1+\gamma)(1-\eta(\tilde{x}^{[j]})^2)^2 & -\gamma(1+\gamma)(1-\eta(\tilde{x}^{[j]})^2)^2 & \gamma^2(1-\eta(\tilde{x}^{[j]})^2)^2 \\
(1+\gamma)(1-\eta(\tilde{x}^{[j]})^2) & 0 & -\gamma(1-\eta(\tilde{x}^{[j]})^2) & 0 \\
(1+\gamma)(1-\eta(\tilde{x}^{[j]})^2) & -\gamma(1-\eta(\tilde{x}^{[j]})^2) & 0 & 0 \\
1 & 0 & 0 & 0
\end{pmatrix}
$$
$$
= \begin{pmatrix}
(1+\gamma)^2 \mathcal{A}^{[j]} & -\gamma(1+\gamma)\mathcal{A}^{[j]} & -\gamma(1+\gamma)\mathcal{A}^{[j]} & \gamma^2 \mathcal{A}^{[j]} \\
(1+\gamma)(1-\eta\sigma_j^2) & 0 & -\gamma(1-\eta\sigma_j^2) & 0 \\
(1+\gamma)(1-\eta\sigma_j^2) & -\gamma(1-\eta\sigma_j^2) & 0 & 0 \\
1 & 0 & 0 & 0
\end{pmatrix}
$$

where $\mathcal{A}^{[j]} := (1 - \eta\sigma_j^2)^2 + 5(\eta\sigma_j^2)^2$.

For the ease of analysis, we define $u := 1 - \gamma \in (0, 1]$ and $t_j := \eta\sigma_j^2, j = 1, 2$. Without loss of generality, we assume $\sigma_1^2 = 1$ in this section. In this case, $t_1 = \eta$ and $t_2 = \eta/\kappa$, where $\kappa$ is the condition number.

Elementary analysis gives the eigenvalues of $\mathcal{B}^{[j]}$:

$$\lambda_1 = (1-u)(1-t_j),$$

$$\lambda_2 = T_0 - \frac{2^{1/3}}{3}\frac{T_1}{\left(T_2 + \sqrt{T_2^2 + 4T_1^3}\right)^{1/3}} + \frac{1}{3 \cdot 2^{1/3}}\left(T_2 + \sqrt{T_2^2 + 4T_1^3}\right)^{1/3},$$

$$\lambda_3 = T_0 + (1 - i\sqrt{3})\frac{2^{1/3}}{6}\frac{T_1}{\left(T_2 + \sqrt{T_2^2 + 4T_1^3}\right)^{1/3}} + (1 + i\sqrt{3})\frac{1}{6 \cdot 2^{1/3}}\left(T_2 + \sqrt{T_2^2 + 4T_1^3}\right)^{1/3},$$

$$\lambda_4 = T_0 + (1 + i\sqrt{3})\frac{2^{1/3}}{6}\frac{T_1}{\left(T_2 + \sqrt{T_2^2 + 4T_1^3}\right)^{1/3}} + (1 - i\sqrt{3})\frac{1}{6 \cdot 2^{1/3}}\left(T_2 + \sqrt{T_2^2 + 4T_1^3}\right)^{1/3},$$

where

$$T_0 = 1 - u + \frac{u^2}{3} - \frac{7}{3}t_j + \frac{7}{3}ut_j - \frac{2}{3}u^2 t_j,$$

$$T_1 = -(-3 + 3u - u^2 + 7t_j - 7ut_j + 2u^2 t_j)^2 + 3(3 - 6u + 4u^2 - u^3 - 10t_j + 20ut_j - 13u^2 t_j + 3u^3 t_j),$$

$$\begin{aligned}T_2 = {} & 9u^4 - 9u^5 + 2u^6 - 72u^2 t_j + 144u^3 t_j - 141u^4 t_j + 69u^5 t_j - 12u^6 t_j + 252t_j^2 - 756ut_j^2 \\ & + 1185u^2 t_j^2 - 1110u^3 t_j^2 + 615u^4 t_j^2 - 186u^5 t_j^2 + 24u^6 t_j^2 - 686t_j^3 + 2058ut_j^3 - 2646u^2 t_j^3 \\ & + 1862u^3 t_j^3 - 756u^4 t_j^3 + 168u^5 t_j^3 - 16u^6 t_j^3.\end{aligned}$$

*Proof idea.* For the two-dimensional component decoupled data, we have

$$\mathbb{E}\left[f(\mathbf{w}_t)\right] - f(\mathbf{w}^*) \geq \sigma_2^2 \mathbb{E}\left[\|\mathbf{w}_t - \mathbf{w}^*\|^2\right] = \sigma_2^2 \mathbb{E}\left[\|\boldsymbol{\epsilon}_t\|^2\right]. \tag{30}$$

By definition of $\Phi$ in Eq.27, we can see that the convergence rate is lower bounded by the convergence rates of the sequences $\{\|\Phi_t^{[j]}\|\}_t$. By the relation Eq.28, we have that the convergence rate of the sequence $\{\|\Phi_t\|\}_t$ is controlled by the magnitude of the top eigenvalue $\lambda_{max}$ of $\mathcal{B}$, if $\Phi_t$ has non-zero component along the eigenvector of $\mathcal{B}$ with eigenvalue $\lambda_{max}(\mathcal{B})$. Specifically, if $|\lambda_{max}| > 1$, $\|\Phi_t^{[j]}\|$ grows at a rate of $|\lambda_{max}|^t$, indicating the divergence of SGD+Nesterov; if $|\lambda_{max}| < 1$, then $\|\Phi_t^{[j]}\|$ converges at a rate of $|\lambda_{max}|^t$.

In the following, We use the eigen-systems of matrices $\mathcal{B}^{[j]}$, especially the top eigenvalue, to analyze the convergence behavior of SGD+Nesterov with any hyper-parameter setting. We show that, for any choice of hyper-parameters (i.e., step-size and momentum parameter), at least one of the following statements must holds:

- $\mathcal{B}^{[1]}$ has an eigenvalue larger than 1.
- $\mathcal{B}^{[2]}$ has an eigenvalue of magnitude $1 - O(1/\kappa)$.

This is formalized in the following two lemmas.

**Lemma 1.** *For any $u \in (0, 1]$, if step size*

$$\eta > \eta_0(u) := \frac{-3 - 2u + 2u^2 + \sqrt{9 + 84u - 164u^2 + 100u^3 - 20u^4}}{2(9 - 15u + 6u^2)}, \tag{31}$$

*then, $\mathcal{B}^{[1]}$ has an eigenvalue larger than 1.*

We will analyze the dependence of the eigenvalues on $\kappa$, when $\kappa$ is large to obtain

**Lemma 2.** *For any $u \in (0, 1]$, if step size*

$$\eta \leq \eta_0(u) := \frac{-3 - 2u + 2u^2 + \sqrt{9 + 84u - 164u^2 + 100u^3 - 20u^4}}{2(9 - 15u + 6u^2)}, \tag{32}$$

*then, $\mathcal{B}^{[2]}$ has an eigenvalue of magnitude $1 - O(1/\kappa)$.*

Finally, we show that $\Phi_t$ has non-zero component along the eigenvector of $\mathcal{B}$ with eigenvalue $\lambda_{max}$, hence the convergence of SGD+Nesterov is controlled by the eigenvalue of $\mathcal{B}^{[j]}$ with the largest magnitude.

**Lemma 3.** *Assume SGD+Nesterov is initialized with $\mathbf{w}_0$ such that both components $\langle \mathbf{w}_0 - \mathbf{w}^*, \mathbf{e}_1 \rangle$ and $\langle \mathbf{w}_0 - \mathbf{w}^*, \mathbf{e}_2 \rangle$ are non-zero. Then, for all $t > 2$, $\Phi_t^{[j]}$ has a non-zero component in the eigen direction of $\mathcal{B}^{[j]}$ that corresponds to the eigenvalue with largest magnitude.*

**Remark 8.** When $\mathbf{w}$ is randomly initialized, the conditions $\langle \mathbf{w}_0 - \mathbf{w}^*, \mathbf{e}_1 \rangle \neq 0$ and $\langle \mathbf{w}_0 - \mathbf{w}^*, \mathbf{e}_2 \rangle \neq 0$ are satisfied with probability 1, since complementary cases form a lower dimensional manifold which has measure 0.

By combining Lemma 1, 2 and 3, we have that SGD+Nesterov either diverges or converges at a rate of $(1 - O(1/\kappa))^t$, and hence, we conclude the non-acceleration of SGD+Nesterov. In addition, Corollary 1 is a special case of Theorem 1 and is proven by combining Lemma 1 and 3.

In high level, the proof ideas of Lemma 1 and 3 is analogous to those of (9), which proves the non-acceleration of stochastic Heavy Ball method over SGD. But the proof idea of Lemma 2 is unique.

*Proof of Lemma 1.* The characteristic polynomial of $B_j$, $j = 1, 2$, are:

$$
\begin{aligned}
D_j(\lambda) &= \lambda^4 - (1+\gamma)^2 \mathcal{A}_j \lambda^3 + \left[ 2\gamma(1+\gamma)^2(1 - \eta\sigma_j^2)\mathcal{A}_j - \gamma^2(1 - \eta\sigma_j^2)^2 - \gamma^2 \mathcal{A}_j \right] \lambda^2 \\
&\quad - \gamma^2(1+\gamma)^2(1 - \eta\sigma_j^2)^2 \mathcal{A}_j \lambda + \gamma^4(1 - \eta\sigma_j^2)^2 \mathcal{A}_j
\end{aligned} \tag{33}
$$

First note that $\lim_{\lambda \to \infty} D_j(\lambda) = +\infty > 0$. In order to show $\mathcal{B}_1$ has an eigenvalue larger than 1, it suffices to verity that $D_1(\lambda = 1) < 0$, because $D_j(\lambda)$ is continuous.

Replacing $\gamma$ by $1 - u$ and $\eta\sigma_1^2$ by $t_1$, we have

$$
\begin{aligned}
D_1(1) &= 4u^2 t_1 - 2u^3 t_1 - 2ut_1^2 - 10u^2 t_1^2 + 6u^3 t_1^2 - 6t_1^3 - 16ut_1^3 + 38u^2 t_1^3 - 16u^3 t_1^3 - 18t_1^4 \\
&\quad + 48ut_1^4 - 42u^2 t_1^4 + 12u^3 t_1^4.
\end{aligned} \tag{34}
$$

Solving for the inequality $D_1(1) < 0$, we have

$$
\eta = t_1 > \frac{-3 - 2u + 2u^2 + \sqrt{9 + 84u - 164u^2 + 100u^3 - 20u^4}}{2(9 - 15u + 6u^2)},
$$

for positive step size $\eta$. $\qquad \square$

*Proof of Lemma 2.* We will show that at least one of the eigenvalues of $\mathcal{B}^{[2]}$ is $1 - O(1/\kappa)$, under the condition in Eq.32.

First, we note that $t_2 = t_1/\kappa = \eta/\kappa$, which is $O(1/\kappa)$. We consider the following cases separately: 1) $u$ is $\Theta(t_2^{0.5})$; 2) $u$ is $o(t_2^{0.5})$ and $\omega(t_2)$; 3) $u$ is $O(t_2)$; 4) $u$ is $\omega(t_2^{0.5})$ and $o(1)$; and 5) $u$ is $\Theta(1)$, the last of which includes the case where momentum parameter is a constant.

Note that, for cases 1-4, $u$ is $o(1)$. In such cases, the step size condition Eq.32 can be written as

$$
\eta < \frac{2}{3}u + o(u). \tag{35}
$$

It is interesting to note that $\eta$ must be $o(1)$ to not diverge, when $u$ is $o(1)$. This is very different to SGD where a constant step size can result in convergence, see (12).

*Case 1: $u$ is $\Theta(t_2^{0.5})$.* In this case, the terms $u^6$, $u^4 t_2$, $u^2 t_2^2$ and $t_2^3$ are of the same order.

We find that

$$
\begin{aligned}
T_0 &= 1 - u + O(t_2), \\
T_1 &= -3u^2 + 12t_2 + O(t_2^{3/2}), \\
T_2 &= 9u^4 - 72u^2 t_2 + 252t_2^2 + O(t_2^{5/2}).
\end{aligned}
$$

Hence,

$$\lambda_1 = 1 - u - t_2 + ut_2$$

$$\lambda_2 = 1 - u - \frac{2^{1/3}}{3}\frac{12t_2 - 3u^2}{(108(4t_2 - u^2)^3)^{1/6}} + \frac{1}{3 \cdot 2^{1/3}}(108(4t_2 - u^2)^3)^{1/6} + O(t_2)$$

$$\lambda_3 = 1 - u + (1 - i\sqrt{3})\frac{2^{1/3}}{6}\frac{12t_2 - 3u^2}{(108(4t_2 - u^2)^3)^{1/6}} + (1 + i\sqrt{3})\frac{1}{6 \cdot 2^{1/3}}(108(4t_2 - u^2)^3)^{1/6} + O(t_2)$$

$$\lambda_4 = 1 - u + (1 + i\sqrt{3})\frac{2^{1/3}}{6}\frac{12t_2 - 3u^2}{(108(4t_2 - u^2)^3)^{1/6}} + (1 - i\sqrt{3})\frac{1}{6 \cdot 2^{1/3}}(108(4t_2 - u^2)^3)^{1/6} + O(t_2)$$

Write $t_2 = cu^2$ asymptotically for some constant $c$. If $4t_2 \geq u^2$, i.e., $4c - 1 \geq 0$, then

$$\lambda_2 = 1 - u + O(t_2),$$

$$\lambda_3 = 1 - u + \frac{\sqrt{4c - 1}}{\sqrt{3}}u + O(t_2),$$

$$\lambda_4 = 1 - u + \frac{\sqrt{4c - 1}}{\sqrt{3}}u + O(t_2).$$

If $4t_2 \leq u^2$, i.e., $4c - 1 \leq 0$, then

$$\lambda_2 = 1 - u + O(t_2),$$

$$\lambda_3 = 1 - u + i\frac{\sqrt{1 - 4c}}{\sqrt{3}}u + O(t_2),$$

$$\lambda_4 = 1 - u + i\frac{\sqrt{1 - 4c}}{\sqrt{3}}u + O(t_2).$$

In either case, the first-order term is of order $u$.

Recall that $t_2 = \eta/\kappa$, then we have

$$u^2 = ct_2 = c\eta/\kappa < \frac{2cu}{3\kappa} \leq \frac{cu}{\kappa} \tag{36}$$

hence, $u < c/\kappa = O(1/\kappa)$. Therefore, $\lambda_i, i = 1, 2, 3, 4$ are $1 - O(1/\kappa)$.

*Case 2: $u$ is $o(t_2^{0.5})$ and $\omega(t_2)$.* In this case,

$$T_0 = 1 - u - 7t_2/3 + o(t_2),$$
$$T_1 = 12t_2 + o(t_2),$$
$$T_2 = 252t_2^2 - 72u^2t_2 + o(t_2^2).$$

Hence,

$$\lambda_1 = 1 - u - t_2 + ut_2,$$
$$\lambda_2 = 1 - u + O(t_2),$$
$$\lambda_3 = 1 - u + 2i\sqrt{t_2} + O(t_2),$$
$$\lambda_4 = 1 - u - 2i\sqrt{t_2} + O(t_2).$$

Assume $u \sim t_2^\alpha$. When $\alpha \in (0.5, 1)$, note that $t_2 = \eta/\kappa \leq \frac{2u}{3\kappa} = O(t_2^\alpha/\kappa)$, we have $t_2^{1-\alpha} = O(1/\kappa)$, hence $t_2 = o(1/\kappa^2)$. This implies that $t_2^{0.5} = o(1/\kappa)$ and $u = o(1/\kappa)$. Therefore, all the eigenvalues $\lambda_i$ are of order $1 - O(1/\kappa)$.

*Case 3: $u$ is $O(t_2)$.* This case if forbidden by the assumption of this lemma. This is because, $t_2 = \eta/\kappa \leq \frac{2u}{3\kappa} = o(u)$ ($1/\kappa$ is $o(1)$). This is contradictory to $u$ is $O(t_2)$.

*Case 4: $u$ is $\Omega(t_2^{0.5})$ and $o(1)$.* In this case, we first consider the terms independent of $t_2$, i.e., constant term and $u$-only terms. These terms can be obtained by putting $t_2 = 0$. In such a setting, the eigenvalues are simplified as:

$$\lambda_1|_{t_2=0} = 1 - u, \quad \lambda_2|_{t_2=0} = 1, \quad \lambda_3|_{t_2=0} = 1 - 2u + u^2, \quad \lambda_4|_{t_2=0} = 1 - u. \tag{37}$$

Note that the $u$-only terms cancel in $\lambda_2$, so the first order after the constant term must be of $t_2$ (could be $t_2/u^2, t_2/u$ etc.). In the following we are going to analyze the $t_2$ terms.

Since $u$ is $\Omega(t_2^{0.5})$, $u^2$ has lower order than $t_2$, and $t_2/u^2$ is $o(1)$. This allows us to do Taylor expansion:

$$
\begin{aligned}
T_1 &= -3u^2(1 - 4\frac{t_2}{u^2}) + f(u) + \text{higher order terms}, \\
T_2 &= 9u^4(1 - 8\frac{t_2}{u^2}) + g(u) + \text{higher order terms},
\end{aligned}
$$

where $f(u)$ and $g(u)$ are $u$-terms only, which, by the above analysis in Eq.37, are shown to contribute nothing to $\lambda_2$. Hence, we use the first terms of $T_1$ and $T_2$ above to analyze the first order term of $\lambda_2$. Plugging in these term to the expression of $\lambda_2$, and keeping the lowest order of $t_2$, we find a zero coefficient of the lowest order $t_2$-term: $t_2/u^2$.

Hence, $\lambda_2$ can be written as:

$$
\lambda_2 = 1 - c\frac{t_2}{u} + O(t_2), \tag{38}
$$

where $c$ is the coefficient.

On the other hand, by Eq.35 and $t_2 = \eta/\kappa$, we have

$$
\frac{t_2}{u} = O(1/\kappa). \tag{39}
$$

Therefore, we can write Eq.38 as $\lambda_2 = 1 - O(1/\kappa)$.

*Case 5: $u$ is $O(1)$.* This is the case where the momentum parameter is $\kappa$-independent. Using the same argument as in case 4, we have zero $u$-only terms. Then, directly taking Taylor expansion with respect to $t_2$ results in:

$$
\lambda_2 = 1 - O(t_2) = 1 - O(1/\kappa). \tag{40}
$$

$\square$

*Proof of Lemma 3.* This proof follows the idea of the proof for stochastic Heavy Ball method in (9). The idea is to examine the subspace spanned by $\Phi_t^{[j]}, t = 0, 1, 2, \cdots, j = 1, 2$, and to prove that the eigenvector of $\mathcal{B}^{[j]}$ corresponding to top eigenvalue (i.e., eigenvalue with largest magnitude) is not orthogonal to this spanned subspace. This in turn implies that there exists a non-zero component of $\Phi_t^{[j]}$ in the eigen direction of $\mathcal{B}^{[j]}$ corresponding to top eigenvalue, and this decays/grows at a rate of $\lambda_{max}^t(\mathcal{B}^{[j]})$.

Recall that $\Phi_{t+1}^{[j]} = \mathcal{B}^{[j]}\Phi_t^{[j]}$. Hence, if $\Phi_t^{[j]}$ has non-zero component in the eigen direction of $\mathcal{B}^{[j]}$ with top eigenvalue, then $\Phi_t^{[j]}$ should also have non-zero component in the same direction. Thus, it suffices to show that at least one of $\Phi_0^{[j]}, \Phi_1^{[j]}, \Phi_2^{[j]}$ and $\Phi_3^{[j]}$ has non-zero component in the eigen direction with top eigenvalue.

Since $\tilde{H}$ is diagonal for this two-dimensional decoupled data, $w^{[1]}$ and $w^{[2]}$ evolves independently, and we can analyze each component separately. In addition, it can be seen that each of the initial values $w_0^{[j]} - (w^*)^{[j]}$, which is non-zero by the assumption of this lemma, just acts as a scale factor during the training. Hence, without loss of generality, we can assume $w_0^{[j]} - (w^*)^{[j]} = 1$, for each $j$. Then, according to the recurrence relation of SGD+Nesterov in Eq.26,

$$
\begin{pmatrix} \epsilon_0^{[j]} \\ \epsilon_{-1}^{[j]} \end{pmatrix} = \begin{pmatrix} 1 \\ 1 \end{pmatrix}, \quad \begin{pmatrix} \epsilon_1^{[j]} \\ \epsilon_0^{[j]} \end{pmatrix} = \begin{pmatrix} s_1^{[j]} \\ 1 \end{pmatrix}, \quad \begin{pmatrix} \epsilon_2^{[j]} \\ \epsilon_1^{[j]} \end{pmatrix} = \begin{pmatrix} (1+\gamma)s_1^{[j]}s_2^{[j]} - \gamma s_2^{[j]} \\ s_1^{[j]} \end{pmatrix},
$$

$$
\begin{pmatrix} \epsilon_3^{[j]} \\ \epsilon_2^{[j]} \end{pmatrix} = \begin{pmatrix} s_3^{[j]}\left[(1+\gamma)^2 s_1^{[j]}s_2^{[j]} - \gamma(1+\gamma)s_2^{[j]}\right] - \gamma s_3^{[j]}\left[(1+\gamma)s_1^{[j]}s_2^{[j]} - \gamma s_2^{[j]}\right] \\ (1+\gamma)s_1^{[j]}s_2^{[j]} - \gamma s_2^{[j]} \end{pmatrix}, \tag{41}
$$

where $s_t^{[j]} = 1 - \eta(\tilde{x}_t^{[j]})^2, t = 1, 2, 3, \cdots$.

Denote the vectorized form of $\Phi_t^{[j]}$ as $\text{vec}(\Phi_t^{[j]})$, which is a $4 \times 1$ column vector. We stack the vectorized forms of $\Phi_0^{[j]}, \Phi_1^{[j]}, \Phi_2^{[j]}, \Phi_3^{[j]}$ to make a $4 \times 4$ matrix, denoted as $\mathcal{M}^{[j]}$:

$$\mathcal{M}^{[j]} = [\text{vec}(\Phi_0^{[j]}) \ \ \text{vec}(\Phi_1^{[j]}) \ \ \text{vec}(\Phi_2^{[j]}) \ \ \text{vec}(\Phi_3^{[j]})]. \tag{42}$$

Note that $\Phi_t^{[j]}, t = 0, 1, 2, 3$, are symmetric tensors, which implies that $\mathcal{M}^{[j]}$ contains two identical rows. Specifically, the second and third row of $\mathcal{M}^{[j]}$ are identical. Therefore, the vector $\mathbf{v} = (0, -1/\sqrt{2}, 1/\sqrt{2}, 0)^T$ is an eigenvector of $\mathcal{M}^{[j]}$ with eigenvalue 0. In fact, $\mathbf{v}$ is also an eigenvector of $\mathcal{B}^{[j]}$ with eigenvalue $\gamma(1 - \eta\sigma_j^2)$. Note that

$$\det(\mathcal{B}^{[j]}) = \gamma^4(1 - \eta\sigma_j^2)^2 \left((1 - \eta\sigma_j^2)^2 + 5(\eta\sigma_j^2)^2\right) \geq \gamma^4(1 - \eta\sigma_j^2)^4.$$

Hence, $\mathbf{v}$ is not the eigenvector along top eigenvalue, and therefore, is orthogonal to the eigen space with top eigenvalue.

In order to prove at least one of $\Phi_t^{[j]}$, $t = 0, 1, 2, 3$, has a non-zero component along the eigen direction of top eigenvalue, it suffices to verify that $\mathcal{M}^{[j]}$ is rank 3, i.e., spans a three-dimensional space. Equivalently, we consider the following matrix

$$\mathcal{M}' := \mathbb{E} \begin{pmatrix} \epsilon_{0,1}^2 & \epsilon_{1,1}^2 & \epsilon_{2,1}^2 \\ \epsilon_{0,1}\epsilon_{-1,1} & \epsilon_{0,1}\epsilon_{1,1} & \epsilon_{1,1}\epsilon_{2,1} \\ \epsilon_{-1,1}^2 & \epsilon_{0,1}^2 & \epsilon_{1,1}^2 \end{pmatrix}, \tag{43}$$

where we omitted the superscript $[j]$ for simplicity of the expression. If the determinant of $\mathcal{M}'^{[j]}$ is not zero, then it is full rank, and hence $\mathcal{M}^{[j]}$ spans a three-dimensional space.

Plug in the expressions in Eq.41, then we have

$$\det(\mathcal{M}'^{[j]}) = \frac{(1-u)t_j^3}{2}(36t_j^2 - 18ut_j^2 + 6ut_j + 3t_j - 3u + 1), \tag{44}$$

where $t_j = \eta\sigma_j^2$ and $u = 1 - \gamma$. $\det(\mathcal{M}'^{[j]}) = 0$ if and only if the polynomial $36t_j^2 - 18ut_j^2 + 6ut_j + 3t_j - 3u + 1 = 0$. Solving for this equation, we have

$$t_j = \frac{-1 - 2u \pm \sqrt{(1 + 2u)^2 + 8(3 - u)(u - 2)}}{12(2 - u)}. \tag{45}$$

We note that, for all $u \in [0, 1)$ both $t_j$ are not positive. This means that, for all $u$ and positive $t_j$, the determinant $\det(\mathcal{M}'^{[j]})$ can never be zero.

Therefore, for each $j = 1, 2$, $\mathcal{M}'^{[j]}$ is full rank, and $\mathcal{M}^{[j]}$ spans a three-dimensional space, which includes the eigenvector with the top eigenvalue of $\mathcal{B}^{[j]}$. Hence, at least one of $\Phi_t^{[j]}, t \in \{0, 1, 2, 3\}$ has non-zero component in the eigen direction with top eigenvalue. By $\Phi_{t+1}^{[j]} = \mathcal{B}^{[j]}\Phi_t^{[j]}$, all succeeding $\Phi_t^{[j]}$ also have non-zero component in the eigen direction with top eigenvalue of $\mathcal{B}^{[j]}$. $\qquad\square$

*Proof of Theorem 1.* Lemma 1 and 2 show that, for any hyper-parameter setting $(\eta, \gamma)$ with $\eta > 0$ and $\gamma \in (0, 1)$, either top eigenvalue of $\mathcal{B}^{[1]}$ is larger than 1 or top eigenvalue of $\mathcal{B}^{[2]}$ is $1 - O(1/\kappa)$. Hence, $|\lambda_{max}|$ is either greater than 1 or is $1 - O(1/\kappa)$. Lemma 3 shows that $\Phi_t$ has non-zero component along the eigenvector of $\mathcal{B}$ with eigenvalue $\lambda_{max}(\mathcal{B})$.

By Eq.28 and Lemma 1 and 2, the sequence $\{\|\Phi_t\|\}_t$ either diverges or converges at a rate of $(1 - O(1/\kappa)^t$. By definition of $\Phi$ in Eq.27, we have that $\{\|\epsilon_t\|\}_t$ either diverges or converges at a rate of $(1 - O(1/\kappa))^t$.

Note that, for the two-dimensional component decoupled data, we have

$$\mathbb{E}[f(\mathbf{w}_t)] - f(\mathbf{w}^*) \geq \sigma_2^2 \mathbb{E}\left[\|\mathbf{w}_t - \mathbf{w}^*\|^2\right] = \sigma_2^2 \mathbb{E}\left[\|\epsilon_t\|^2\right]. \tag{46}$$

Therefore, the convergence rate of SGD+Nesterov is lower bounded by $(1 - O(1/\kappa))^t$. Note that the convergence rate of SGD on this data is $(1 - O(1/\kappa))^t$, hence SGD+Nesterov does not accelerate SGD on the two-dimensional component decoupled dataset. $\qquad\square$

*Proof of Corollary 1.* When $\eta = 1/L_1$ and $\gamma \in [0.6, 1]$, the condition in Eq.31 is satisfied. By Lemma 1, the top eigenvalue $\lambda_{max}^{[1]}$ of $\mathcal{B}^{[1]}$ is larger than 1. By Lemma 3, $\Phi^{[1]}$ has non-zero component along the eigenvector with this top eigenvalue $\lambda_{max}^{[1]}$. Hence, $|\langle \mathbf{w}_t - \mathbf{w}^*, \mathbf{e}_1 \rangle|$ grows at a rate of $(\lambda_{max}^{[1]})^t$. □

## D  PROOF OF THEOREM 2

We first give a lemma that is useful for dealing with the mini-batch scenario:

**Lemma 4.** *If square loss $f$ is in the interpolated setting, i.e., there exists $\mathbf{w}^*$ such that $f(\mathbf{w}^*) = 0$, then $\mathbb{E}[\tilde{H}_m H^{-1} \tilde{H}_m] - H \preceq \frac{1}{m}(\tilde{\kappa} - 1) H$.*

*Proof.*

$$\mathbb{E}[\tilde{H}_m H^{-1} \tilde{H}_m] = \frac{1}{m^2}\mathbb{E}\left[\sum_{i=1}^{m} \tilde{\mathbf{x}}_i \tilde{\mathbf{x}}_i^T H^{-1} \tilde{\mathbf{x}}_i \tilde{\mathbf{x}}_i^T + \sum_{i \neq j} \tilde{\mathbf{x}}_i \tilde{\mathbf{x}}_i^T H^{-1} \tilde{\mathbf{x}}_j \tilde{\mathbf{x}}_j^T\right] \preceq \frac{1}{m}\tilde{\kappa}H + \frac{m-1}{m}H.$$

□

*Proof of Theorem 2.* By Eq.8, we have

$$\|\mathbf{v}_{t+1} - \mathbf{w}^*\|_{H^{-1}}^2 = \underbrace{\|(1-\alpha)\mathbf{v}_t + \alpha\mathbf{u}_t - \mathbf{w}^*\|_{H^{-1}}^2}_{\tilde{A}} + \underbrace{\delta^2\|\tilde{H}_m(\mathbf{u}_t - \mathbf{w}^*)\|_{H^{-1}}^2}_{\tilde{B}}$$
$$\underbrace{-2\delta\langle \tilde{H}_m(\mathbf{u}_t - \mathbf{w}^*), (1-\alpha)\mathbf{v}_t + \alpha\mathbf{u}_t - \mathbf{w}^*\rangle_{H^{-1}}}_{\tilde{C}}$$

Using the convexity of the norm $\|\cdot\|_{H^{-1}}$ and the fact that $\mu$ is the smallest non-zero eigenvalue of the Hessian, we get

$$\mathbb{E}[\tilde{A}] \leq (1-\alpha)\|\mathbf{v}_t - \mathbf{w}^*\|_{H^{-1}}^2 + \alpha\|\mathbf{u}_t - \mathbf{w}^*\|_{H^{-1}}^2 \leq (1-\alpha)\|\mathbf{v}_t - \mathbf{w}^*\|_{H^{-1}}^2 + \frac{\alpha}{\mu}\|\mathbf{u}_t - \mathbf{w}^*\|^2.$$

Applying Lemma 4 on the term $\tilde{B}$, we have

$$\mathbb{E}[\tilde{B}] \leq \delta^2\tilde{\kappa}_m\|\mathbf{u}_t - \mathbf{w}^*\|_H^2. \tag{47}$$

Note that $\mathbb{E}[\tilde{H}_m] = H$, then

$$\begin{aligned}
\mathbb{E}[\tilde{C}] &= -2\delta\langle \mathbf{u}_t - \mathbf{w}^*, (1-\alpha)\mathbf{v}_t + \alpha\mathbf{u}_t - \mathbf{w}^*\rangle \\
&= -2\delta\langle \mathbf{u}_t - \mathbf{w}^*, \mathbf{u}_t - \mathbf{w}^* + \frac{1-\alpha}{\alpha}(\mathbf{u}_t - \mathbf{w}_t)\rangle \\
&= -2\delta\|\mathbf{u}_t - \mathbf{w}^*\|^2 + \frac{1-\alpha}{\alpha}\delta\left(\|\mathbf{w}_t - \mathbf{w}^*\|^2 - \|\mathbf{u}_t - \mathbf{w}^*\|^2 - \|\mathbf{w}_t - \mathbf{u}_t\|^2\right) \\
&\leq \frac{1-\alpha}{\alpha}\delta\|\mathbf{w}_t - \mathbf{w}^*\|^2 - \frac{1+\alpha}{\alpha}\delta\|\mathbf{u}_t - \mathbf{w}^*\|^2. \tag{48}
\end{aligned}$$

Therefore,

$$\mathbb{E}\left[\|\mathbf{v}_{t+1} - \mathbf{w}^*\|_{H^{-1}}^2\right] \leq (1-\alpha)\|\mathbf{v}_t - \mathbf{w}^*\|_{H^{-1}}^2 + \frac{1-\alpha}{\alpha}\delta\|\mathbf{w}_t - \mathbf{w}^*\|^2 + \left(\frac{\alpha}{\mu} - \frac{1+\alpha}{\alpha}\delta\right)\|\mathbf{u}_t - \mathbf{w}^*\|^2 + \delta^2\tilde{\kappa}_m\|\mathbf{u}_t - \mathbf{w}^*\|_H^2.$$

On the other hand, by the fact that $\mathbb{E}[\tilde{H}_m^2] \preceq \frac{1}{m}\mathbb{E}[\tilde{H}_1^2] + \frac{m-1}{m}H^2$, (see (12)),

$$\begin{aligned}
\mathbb{E}[\|\mathbf{w}_{t+1} - \mathbf{w}^*\|^2] &= \mathbb{E}\left[\|\mathbf{u}_t - \mathbf{w}^* - \eta\tilde{H}_m(\mathbf{u}_t - \mathbf{w}^*)\|^2\right] \\
&\leq \|\mathbf{u}_t - \mathbf{w}^*\|^2 - 2\eta\|\mathbf{u}_t - \mathbf{w}^*\|_H^2 + \eta^2 L_m\|\mathbf{u}_t - \mathbf{w}^*\|_H^2. \tag{49}
\end{aligned}$$

Hence,

$$\mathbb{E}\left[\|\mathbf{v}_{t+1} - \mathbf{w}^*\|^2_{H^{-1}} + \frac{\delta}{\alpha}\|\mathbf{w}_{t+1} - \mathbf{w}^*\|^2\right]$$

$$\leq \quad (1-\alpha)\|\mathbf{v}_t - \mathbf{w}^*\|^2_{H^{-1}} + \frac{1-\alpha}{\alpha}\delta\|\mathbf{w}_t - \mathbf{w}^*\|^2 + \underbrace{(\alpha/\mu - \delta)}_{c_1}\|\mathbf{u}_t - \mathbf{w}^*\|^2$$

$$+ \underbrace{\left(\delta^2\tilde{\kappa}_m + \delta\eta^2 L_m/\alpha - 2\eta\delta/\alpha\right)}_{c_2}\|\mathbf{u}_t - \mathbf{w}^*\|^2_H.$$

By the condition in Eq.10, $c_1 \leq 0, c_2 \leq 0$, then the last two terms are non-positive. Hence,

$$\mathbb{E}\left[\|\mathbf{v}_{t+1} - x^*\|^2_{H^{-1}} + \frac{\delta}{\alpha}\|\mathbf{w}_{t+1} - \mathbf{w}^*\|^2\right] \quad \leq \quad (1-\alpha)\left(\|\mathbf{v}_t - \mathbf{w}^*\|^2_{H^{-1}} + \frac{\delta}{\alpha}\|\mathbf{w}_t - \mathbf{w}^*\|^2\right)$$

$$\leq \quad (1-\alpha)^{t+1}\left(\|\mathbf{v}_0 - \mathbf{w}^*\|^2_{H^{-1}} + \frac{\delta}{\alpha}\|\mathbf{w}_0 - \mathbf{w}^*\|^2\right).$$

$\square$

## E  PROOF OF THEOREM 4

*Proof.* The update rule for variable $\mathbf{v}$ is, as in Eq.8:

$$\mathbf{v}_{t+1} = (1-\alpha)\mathbf{v}_t + \alpha\mathbf{u}_t - \delta\tilde{\nabla}_m f(\mathbf{u}_t). \tag{50}$$

By $1/\mu$-strong convexity of $g$, we have

$$g(\mathbf{v}_{t+1}) \leq g\left((1-\alpha)\mathbf{v}_t + \alpha\mathbf{u}_t\right) + \langle\nabla g\left((1-\alpha)\mathbf{v}_t + \alpha\mathbf{u}_t\right), -\delta\tilde{\nabla}_m f(\mathbf{u}_t)\rangle + \delta^2\frac{1}{2\mu}\|\tilde{\nabla}_m f(\mathbf{u}_t)\|^2. \tag{51}$$

Taking expectation on both sides, we get

$$\mathbb{E}[g(\mathbf{v}_{t+1})] \quad = \quad g\left((1-\alpha)\mathbf{v}_t + \alpha\mathbf{u}_t\right) + \langle\nabla g\left((1-\alpha)\mathbf{v}_t + \alpha\mathbf{u}_t\right), -\delta\nabla f(\mathbf{u}_t)\rangle + \delta^2\frac{1}{2\mu}\mathbb{E}[\|\tilde{\nabla}_m f(\mathbf{u}_t)\|^2]$$

$$\leq \quad (1-\alpha)g(\mathbf{v}_t) + \alpha g(\mathbf{u}_t) - \delta(1-\epsilon)\langle(1-\alpha)\mathbf{v}_t + \alpha\mathbf{u}_t - \mathbf{w}^*, \mathbf{u}_t - \mathbf{w}^*\rangle + \delta^2\frac{2L_m}{2\mu}f(\mathbf{u}_t),$$

where in the last inequality we used the convexity of $g$ and the assumption in the Theorem.

By Eq.48,

$$-\delta(1-\epsilon)\langle(1-\alpha)\mathbf{v}_t + \alpha\mathbf{u}_t - \mathbf{w}^*, \mathbf{u}_t - \mathbf{w}^*\rangle \leq \frac{\delta(1-\epsilon)}{2}\left(\frac{1-\alpha}{\alpha}\|\mathbf{w}_t - \mathbf{w}^*\|^2 - \frac{1+\alpha}{\alpha}\|\mathbf{u}_t - \mathbf{w}^*\|^2\right).$$

By the strong convexity of $g$,

$$\alpha g(\mathbf{u}_t) \leq \frac{\alpha}{2\mu}\|\mathbf{u}_t - \mathbf{w}^*\|^2.$$

Hence,

$$\mathbb{E}[g(\mathbf{v}_{t+1})] \leq (1-\alpha)g(\mathbf{v}_t) + (1-\alpha)\frac{\delta(1-\epsilon)}{2\alpha}\|\mathbf{w}_t - \mathbf{w}^*\|^2 + \left(\frac{\alpha}{2\mu} - \frac{\delta(1+\alpha)(1-\epsilon)}{2\alpha}\right)\|\mathbf{u}_t - \mathbf{w}^*\|^2 + \delta^2\kappa_m f(\mathbf{u}_t). \tag{52}$$

On the other hand,

$$\mathbb{E}[\|\mathbf{w}_{t+1} - \mathbf{w}^*\|^2] \quad = \quad \|\mathbf{u}_t - \mathbf{w}^*\|^2 - 2\eta\langle\mathbf{u}_t - \mathbf{w}^*, \nabla f(\mathbf{u}_t)\rangle + \eta^2\mathbb{E}[\|\tilde{\nabla}_m f(\mathbf{u}_t)\|^2]$$

$$\leq \quad \|\mathbf{u}_t - \mathbf{w}^*\|^2 - 2\eta f(\mathbf{u}_t) + \eta^2 \cdot 2L_m f(\mathbf{u}_t), \tag{53}$$

where in the last inequality we used the convexity of $f$.

Multiply a factor of $\frac{\delta(1-\epsilon)}{2\alpha}$ onto Eq.53 and add it to Eq.52, then

$$
\begin{aligned}
\mathbb{E}\left[g(\mathbf{v}_{t+1}) + \frac{\delta(1-\epsilon)}{2\alpha}\|\mathbf{w}_{t+1} - \mathbf{w}^*\|^2\right] \leq\ & (1-\alpha)g(\mathbf{v}_t) + (1-\alpha)\frac{\delta(1-\epsilon)}{2\alpha}\|\mathbf{w}_t - \mathbf{w}^*\|^2 \\
& + \frac{1}{2}\left(\frac{\alpha}{\mu} - \delta(1-\epsilon)\right)\|\mathbf{u}_t - \mathbf{w}^*\|^2 \\
& + \left(\frac{\delta}{\alpha}(1-\epsilon)(\eta^2 L_m - \eta) + \delta^2\kappa_m\right)f(\mathbf{u}_t). \quad (54)
\end{aligned}
$$

If the hyper-parameters are selected as:

$$
\eta = \frac{1}{2L_m}, \quad \alpha = \frac{1-\epsilon}{2\kappa_m}, \quad \delta = \frac{1}{2L_m}, \tag{55}
$$

then the last two terms in Eq.54 are non-positive. Hence,

$$
\mathbb{E}\left[g(\mathbf{v}_{t+1}) + \frac{\delta(1-\epsilon)}{2\alpha}\|\mathbf{w}_{t+1} - \mathbf{w}^*\|^2\right] \leq (1-\alpha)\mathbb{E}\left[g(\mathbf{v}_t) + \frac{\delta(1-\epsilon)}{2\alpha}\|\mathbf{w}_t - \mathbf{w}^*\|^2\right], \tag{56}
$$

which implies

$$
\mathbb{E}\left[g(\mathbf{v}_t) + \frac{\delta(1-\epsilon)}{2\alpha}\|\mathbf{w}_t - \mathbf{w}^*\|^2\right] \leq (1 - \frac{1-\epsilon}{2\kappa_m})^t\mathbb{E}\left[g(\mathbf{v}_0) + \frac{\delta(1-\epsilon)}{2\alpha}\|\mathbf{w}_0 - \mathbf{w}^*\|^2\right]. \tag{57}
$$

Since $f(\mathbf{w}_t) \leq L/2 \cdot \|\mathbf{w}_t - \mathbf{w}^*\|^2$ (by smoothness), then we have the final conclusion

$$
\mathbb{E}[f(\mathbf{w}_t)] \leq C \cdot (1 - \frac{1-\epsilon}{2\kappa_m})^t, \tag{58}
$$

with $C$ being a constant. $\qquad\square$

# F  EVALUATION ON SYNTHETIC DATA

## F.1  DISCUSSION ON SYNTHETIC DATASETS

**Component Decoupled Data.**    The data is defined as follows (also defined in section 3):

Fix an arbitrary $\mathbf{w}^* \in \mathbb{R}^2$ and let $z$ be randomly drawn from the zero-mean Gaussian distribution with variance $\mathbb{E}[z^2] = 2$, i.e. $z \sim \mathcal{N}(0, 2)$. The data points $(\mathbf{x}, y) \in \mathcal{D}$ are constructed as follow:

$$
\mathbf{x} = \begin{cases} \sigma_1 \cdot z \cdot \mathbf{e}_1 & \text{w.p. } 0.5 \\ \sigma_2 \cdot z \cdot \mathbf{e}_2 & \text{w.p. } 0.5, \end{cases} \quad \text{and } y = \langle \mathbf{w}^*, \mathbf{x} \rangle, \tag{59}
$$

where $\mathbf{e}_1, \mathbf{e}_2 \in \mathbb{R}^2$ are canonical basis vectors, $\sigma_1 > \sigma_2 > 0$.

Note that the corresponding square loss function on $\mathcal{D}$ is in the interpolation regime, since $f(\mathbf{w}^*) = 0$. The Hessian and stochastic Hessian matrices turn out to be

$$
H = \begin{bmatrix} \sigma_1^2 & 0 \\ 0 & \sigma_2^2 \end{bmatrix}, \quad \tilde{H} = \begin{bmatrix} (x^{[1]})^2 & 0 \\ 0 & (x^{[2]})^2 \end{bmatrix}. \tag{60}
$$

Note that $\tilde{H}$ is diagonal, which implies that stochastic gradient based algorithms applied on this data evolve independently in each coordinate. This allows a simplified directional analysis of the algorithms applied.

Here we list some useful results for our analysis. The fourth-moment of Gaussian variable $\mathbb{E}[z^4] = 3\mathbb{E}[z^2]^2 = 12$. Hence, $\mathbb{E}[(x^{[j]})^2] = \sigma_j^2$ and $\mathbb{E}[(x^{[j]})^4] = 6\sigma_j^4$, where superscript $j = 1, 2$ is the index for coordinates in $\mathbb{R}^2$. It is easy to check that $\kappa_1 = 6\sigma_1^2/\sigma_2^2$ and $\tilde{\kappa} = 6$.

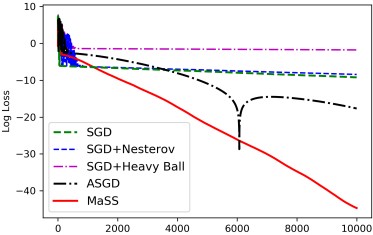 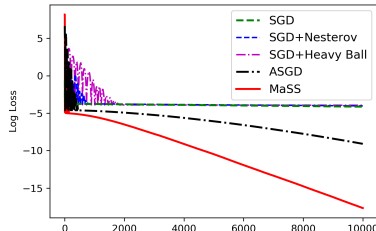

Figure 6: Fast convergence of MaSS and non-acceleration of SGD+Nesterov on component decoupled data. (left) $\sigma_1^2 = 1$ and $\sigma_2^2 = 1/2^1 2$; (right) $\sigma_1^2 = 1$ and $\sigma_2^2 = 1/2^1 5$.

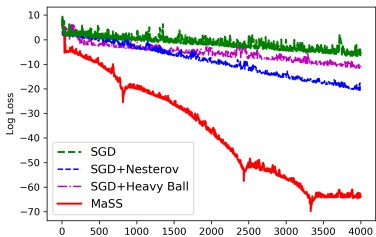 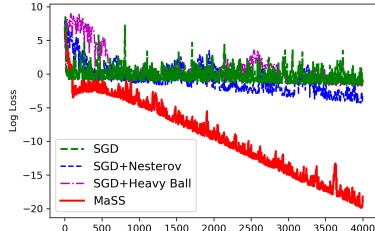

Figure 7: Fast convergence of MaSS and non-acceleration of SGD+Nesterov on 3-d Gaussian data. (left) $\sigma_1^2 = 1$ and $\sigma_2^2 = 1/2^9$; (right) $\sigma_1^2 = 1$ and $\sigma_2^2 = 1/2^1 2$.

**Gaussian Data.** Suppose the data feature vectors $\{\mathbf{x}_i\}$ are zero-mean Gaussian distributed, and $y_i = \langle \mathbf{w}^*, \mathbf{x}_i \rangle$, $\forall i$, where $\mathbf{w}^*$ is fixed but unknown. Then, by the fact that $\mathbb{E}[z_1 z_2 z_3 z_4] = \mathbb{E}[z_1 z_2]\mathbb{E}[z_3 z_4] + \mathbb{E}[z_1 z_3]\mathbb{E}[z_2 z_4] + \mathbb{E}[z_1 z_4]\mathbb{E}[z_2 z_3]$ for zero-mean Gaussian random variables $z_1, z_2, z_3$ and $z_4$, we have

$$
\begin{aligned}
\mathbb{E}[\tilde{H}\tilde{H}] &= (2H + tr(H))H, \\
\mathbb{E}[\tilde{H}H^{-1}\tilde{H}] &= \mathbb{E}[\tilde{\mathbf{x}}\tilde{\mathbf{x}}^T H^{-1}\tilde{\mathbf{x}}\tilde{\mathbf{x}}^T] = (2+d)H,
\end{aligned}
$$

where $d$ is the dimension of the feature vectors. Hence $L_1 = 2\lambda_{max}(H) + tr(H)$ and $\tilde{\kappa} = 2 + d$, and $\kappa_1 = L_1/\mu$. This implies a convergence rate of $O(e^{-t/\sqrt{(2+d)\kappa_1}})$ of MaSS when batch size is 1. Particularly, if the feature vectors are $n$-dimensional, e.g., as in kernel learning, then MaSS with mini batches of size 1 has a convergence rate of $O(e^{-t/\sqrt{n\kappa_1}})$.

### F.2 EVALUATION OF FAST CONVERGENCE OF MASS AND NON-ACCELERATION OF SGD+NESTEROV

In this subsection, we show additional empirical verification for the fast convergence of MaSS, as well as the non-acceleration of SGD+Nesterov, on synthetic data. In addition, we show the divergence of SGD+Nesterov when using the same step size as SGD and MaSS, as indicated by Corollary 1.

We consider two families of synthetic datasets:

- *Component decoupled: (as defined in Section 3).* Fix an arbitrary $\mathbf{w}^* \in \mathbb{R}^2$ with all components non-zero. $\mathbf{x}_i$ is drawn from $\mathcal{N}(\mathbf{0}, diag(2\sigma_1^2, 0))$ or $\mathcal{N}(\mathbf{0}, diag(0, 2\sigma_2^2))$ with probability 0.5 each. $y_i = \langle \mathbf{w}^*, \mathbf{x}_i \rangle$ for all $i$.

- *3-d Gaussian*: Fix an arbitrary $\mathbf{w}^* \in \mathbb{R}^3$ with all components non-zero. $\mathbf{x}_i$ are independently drawn from $\mathcal{N}(\mathbf{0}, diag(\sigma_1^2, \sigma_1^2, \sigma_2^2))$, and $y_i = \langle \mathbf{w}^*, \mathbf{x}_i \rangle$ for all $i$.

**Non-acceleration of SGD+Nesterov and accelerated convergence of MaSS.** We compare MaSS with SGD and SGD+Nesterov on linear regression with the above datasets. Each comparison is

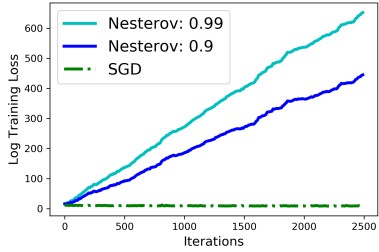

Figure 8: Divergence of SGD+Nesterov with large step size. Step size: $\eta^* = 1/L_1 = 1/6$, and momentum parameter: $\gamma$ is 0.9 or 0.99.

performed on either 3-d Gaussian or component decoupled data with fixed $\sigma_1$ and $\sigma_2$. For each setting of $(\sigma_1, \sigma_2)$, we randomly select $\mathbf{w}^*$, and generate 2000 samples for the dataset. Batch sizes for all algorithms are set to be 1. We report the performances of SGD, SGD+Nesterov and SGD+HB using their best hyper-parameter setting selected from dense grid search. On the other hand, we do not tune hyper-parameters of MaSS, but use the suggested setting by our theoretical analysis, Eq. 14. Specifically, we use

$$\text{Component decoupled:} \quad \eta_1^* = \frac{1}{6}, \quad \eta_2^* = \frac{5}{36 + 6\sigma_2}, \quad \gamma^* = \frac{6 - \sigma_2}{6 + \sigma_2}; \quad (61\text{a})$$

$$\text{3-d Gaussian:} \quad \eta_1^* = \frac{1}{4}, \quad \eta_2^* = \frac{1}{5}\frac{\sqrt{20}}{\sqrt{20} + \sigma_2}, \quad \gamma^* = \frac{\sqrt{20} - \sigma_2}{\sqrt{20} + \sigma_2}. \quad (61\text{b})$$

For ASGD, we use the setting suggested by (8).

Figure 6 (in addition to Fig 4) and Figure 7 show the curves of the compared algorithms under various data settings. We observe that: 1) SGD+Nesterov with its best hyper-parameters is almost identical to the optimal SGD; 2) MaSS, with the suggested hyper-parameter selections, converges faster than all of the other algorithms, especially SGD. These observations are consistent with our theoretical results about non-acceleration of SGD+Nesterov, Theorem 1, and accelerated convergence of MaSS, Theorem 3.

Recall that MaSS differs from SGD+Nesterov by only a compensation term, this experiment illustrates the importance of this term. Note that the vertical axis is log scaled. Then the linear decrease of log losses in the plots implies an exponential loss decrease, and the slopes correspond to the coefficients in the exponents.

**Divergence of SGD+Nesterov with large step size.** As discussed in Corollary 1, SGD+Nesterov diverges with step size $\eta^* = 1/L_1$ (when $\gamma \in [0.6, 1]$), which is the optimal choice of step size for both SGD and MaSS. We run SGD+Nesterov, with step size $\eta^* = 1/L_1$, to optimize the square loss function on component decoupled data mentioned above. Figure 8 shows the divergence of SGD+Nesterov with two common choices of momentum parameter $\gamma$: 0.9 and 0.99.

### F.3 COMPARISON OF VASWANI'S METHOD (21) WITH SGD AND MASS ON GAUSSIAN-DISTRIBUTED DATA

The analysis in Vaswani et. al. (21) is based on the *strong growth condition* (SGC). Assuming SGC with the parameter $\rho$ on the loss function they prove convergence rate $\left(1 - \sqrt{1/\rho^2\kappa}\right)^t$ of their method (called SGD with Nesterov acceleration in their paper), where $t$ is the iteration number. In the following, we show that, on a simple (zero-mean) Gaussian distributed data, this rate is slower than that of SGD, which has a rate of $(1 - 1/\kappa)^t$. On the other hand, MaSS achieves the accelerated rate $\left(1 - 1/\sqrt{(2 + d)\kappa}\right)^t$.

Consider the problem of minimizing the squared loss, $f(\mathbf{w}) = \sum_i f_i(\mathbf{w}) = \frac{1}{2} \sum_i (\mathbf{w}^T \mathbf{x}_i - y_i)^2$, over a zero-mean Gaussian distributed dataset, as defined in F.1. Then,

$$
\begin{aligned}
\mathbb{E}_i \|\nabla f_i(\mathbf{w})\|^2 &= (\mathbf{w} - \mathbf{w}^*)^T \mathbb{E}_i \left[ \mathbf{x}_i \mathbf{x}_i^T \mathbf{x}_i \mathbf{x}_i^T \right] (\mathbf{w} - \mathbf{w}^*) \\
&= (\mathbf{w} - \mathbf{w}^*)^T (2H^2 + tr(H)H)(\mathbf{w} - \mathbf{w}^*),
\end{aligned}
\tag{62}
$$

whereas,

$$
\|\nabla f(\mathbf{w})\|^2 = (\mathbf{w} - \mathbf{w}^*)^T H^2 (\mathbf{w} - \mathbf{w}^*).
\tag{63}
$$

According to the definition of SGC,

$$
\mathbb{E}_i \|\nabla f_i(\mathbf{w})\|^2 \le \rho \|\nabla f(\mathbf{w})\|^2.
\tag{64}
$$

Hence the SGC parameter $\rho$ must satisfy

$$
\rho \ge 2 + \frac{tr(H)}{\lambda_{min}(H)} > \kappa,
\tag{65}
$$

where $\kappa$ is the condition number.

In this case, the convergence rate $\left(1 - \sqrt{1/\rho^2\kappa}\right)^t$ of Vaswani's method would be slower than $\left(1 - 1/\kappa^{3/2}\right)^t$, which is slower than SGD.

On the other hand, MaSS accelerates over SGD on this dataset. Recall from Section F.1 that, for this Gaussian data, $\tilde{\kappa} = 2 + d$, where $d$ is the dimension of the the data. According to Theorem 3, the convergence rate of MaSS is $\left(1 - 1/\sqrt{(2+d)\kappa}\right)^t$, which is faster than that of SGD.

# G  MINI-BATCH DEPENDENCE OF CONVERGENCE SPEED AND OPTIMAL HYPER-PARAMETERS

## G.1  ANALYSIS

The two critical points are defined as follow:

$$
m_1^* = \min(L_1/L, \tilde{\kappa}), m_2^* = \max(L_1/L, \tilde{\kappa}).
$$

When $m < m_1^*$, we have $m < L_1/L$ and $m < \tilde{\kappa}$, which implies $L_m \approx L_1/m$, $\kappa_m \approx \kappa_1/m$ and $\tilde{\kappa}_m \approx \tilde{\kappa}/m$. Plugging into Eq.13, we find that the optimal selection of hyper-parameters is approximated by:

$$
\eta^*(m) \approx m \cdot \eta^*(1), \alpha^*(m) \approx m \cdot \alpha^*(1), \delta^*(m) \approx m \cdot \delta^*(1),
$$

and the convergence rate in Eq.16 is approximately $O(e^{-m \cdot t/\sqrt{\kappa_1 \tilde{\kappa}}})$, the latter of which indicates a linear gains in convergence when $m$ increases.

Now consider the following two cases: (i) For $m \ge L_1/L$, we have $\kappa_m \le 2\kappa$ following the definition of $L_m$ in Eq.2; (ii) For $m \ge \tilde{\kappa}$, $\tilde{\kappa}_m = \tilde{\kappa}/m + (m-1)/m \le 2$. When either case holds $m \ge m_1^*$, the convergence rate (Eq.16) becomes

$$
O\left( \max\left\{ e^{-\frac{\sqrt{m} \cdot t}{\sqrt{2\kappa(\tilde{\kappa}+m-1)}}}, e^{-\frac{\sqrt{m} \cdot t}{\sqrt{2\kappa_1 + 2(m-1)\kappa}}} \right\} \right),
$$

indicating that increasing the mini-batch size results in sublinear gain in the convergence speed, of at most $\sqrt{m}$. Moreover, in the saturation regime $m > m_2^*$ (i.e., when both conditions (i) and (ii) hold), the convergence rate is then upper-bounded by $O(\exp(-\frac{t}{2\sqrt{\kappa}}))$. That implies that a single iteration of MaSS with mini-batch size $m$ is equivalent up to a multiplicative factor of 2 to an iteration with full gradient.

## G.2 EMPIRICAL VERIFICATION

We empirically verify the three-regime partition observed in section 5 using zero-mean Gaussian data.

In this evaluation, we set the covariant matrix of the (zero-mean) Gaussian to be:

$$\Sigma = diag(\underbrace{1, \cdots, 1}_{8}, \underbrace{2^{-10}, \cdots, 2^{-10}}_{40}). \tag{66}$$

and generate 2000 samples following the Gaussian distribution $\mathcal{N}(\mathbf{0}, \Sigma)$. Recall from Example 1 that, for zero-mean Gaussian data, $L_1 = 2\lambda_{max}(H) + tr(H)$ and $\tilde{\kappa} = 2 + d$. In this case, $L_1/L \approx 10$ and $\tilde{\kappa} \approx 50$. The two critical values $m_1^*, m_2^*$ of mini-batch size are then:

$$m_1^* \approx 10, \quad m_2^* \approx 50.$$

In the experiments, we run MaSS with a variaty of mini-batch size $m$, ranging from 1 to 160, on this Gaussian dataset. For each training process, we compute the convergence speed $s(m)$, which is defined to be the inverse of the number of iterations needed to achieve a training error of $\varepsilon$. Running 1 iteration of MaSS with mini-batch of size $m$ is almost as effective as running $s(m)/s(1)$ iterations of MaSS with mini-batch of size 1.

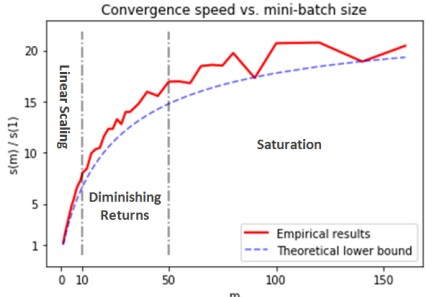

Figure 9 demonstrates the convergence speed $s(m)$ as a function of the mini-batch size $m$. We see that the three regimes defined by $m_1^*$ and $m_2^*$ coincide with the analysis in subsection 5: left ($m < m_1^*$), almost linear; middle ($m \in [m_1^*, m_2^*]$), sublinear; right ($m > m_2^*$), saturation.

Figure 9: Convergence speed per iteration as a function of mini-batch size $m$. Red solid curve: experimental results. Larger $s(m)$ indicates faster convergence. Blue dash curve: theoretical lower bound $\sqrt{\kappa_m \tilde{\kappa}_m}$. Critical mini-batch size values: $m_1^* \approx 10, m_2^* \approx 50$.

## H EXPERIMENTAL SETUP

### H.1 NEURAL NETWORK ARCHITECTURES

**Fully-connected Network.** The fully-connected neural network has 3 hidden layers, with 100 ReLU-activated neurons in each layer. After each hidden layer, there is a dropout layer with keep probability 0.5. This network takes 784-dimensional vectors as input, and has 10 softmax-activated output neurons. It has ≈99k trainable parameters in total.

**Convolutional Neural Network (CNN).** The CNN we considered has three convolutional layers with kernel size of $5 \times 5$ and without padding. The first two convolutional layers have 64 channels each, while the last one has 128 channels. Each convolutional layer is followed by a $2 \times 2$ max pooling layer with stride of 2. On top of the last max pooling layer, there is a fully-connected ReLU-activated layer of size 128 followed by the output layer of size 10 with softmax non-linearity. A dropout layer with keep probability 0.5 is applied after the full-connected layer. The CNN has ≈576k trainable parameters in total.

**Residual Network (ResNet).** We train a ResNet (7) with 32 convolutional layers. The ResNet-32 has a sequence of 15 residual blocks: the first 5 blocks have an output of shape $32 \times 32 \times 16$, the following 5 blocks have an output of shape $16 \times 16 \times 32$ and the last 5 blocks have an output of shape $8 \times 8 \times 64$. On top of these blocks, there is a $2 \times 2$ average pooling layer with stride of 2, followed by a output layer of size 10 with softmax non-linearity. The ResNet-32 has ≈467k trainable parameters in total.

We use the fully-connected network to classify the MNIST dataset, and use CNN and ResNet to classify the CIFAR-10 dataset.

## H.2 EXPERIMENTAL SETUP FOR TEST PERFORMANCE OF MASS

The 3-layer CNN we use for this experiment is slightly different from the aforementioned one. We insert a batch normalization (BN) layer after each convolution computation, and remove the dropout layer in the fully-connected phase.

**Reduction of Learning Rate.** On both CNN and ResNet-32, we initialize the learning rate of SGD, SGD+Nesterov and Mass using the same value, while that of Adam is set to 0.001, which is common default setting. On CNN, the learning rates for all algorithms drop by 0.1 at 60 and 120 epochs, and we train for total 150 epochs. On ResNet-32, the learning rates drop by 0.1 at 150 and 225 epochs, and we train for total 300 epochs.

Whenever the learning rate of MaSS reduced, we restart the MaSS with the latest learned weights. The reason for restarting is to avoid the mismatch between the large momentum term and small gradient descent update after the reduction of learning rate.

**Data Augmentation.** We augment the Cifar-10 training data by enabling random horizontal flip, random horizontal shift and random vertical shift of the images. For the random shift of images, we allow a shift range of 10% of the image width or height.

**Hyper-parameter Selection for MaSS.** Since the reduction of learning rate may affect the suggested value of $\delta$ (c.f. Eq.13), we consider $(\eta, \alpha, \tilde{\kappa}_m)$ as independent hyper-parameters instead, where $\tilde{\kappa}_m = \eta/(\alpha\delta)$. Observing the fact that $\tilde{\kappa}_m \leq \kappa_m$, we select $\tilde{\kappa}_m$ in the range $(1, 1/\alpha)$. In our experiments, we set $\alpha = 0.05$, corresponding to the "momentum parameter" $\gamma$ being 0.90, which is commonly used in Nesterov's method. We find that good choices of $\tilde{\kappa}_m$ often locate in the interval $[2, 25]$ for mini-batch size $m = 64$ in our experiments.

The classification results of MaSS in Table 6 are obtained under the following hyper-parameter settings:

- **CNN**:
  $\eta = 0.01$ (initial), $\alpha = 0.05$, $\tilde{\kappa}_m = 3$;
  $\eta = 0.3$ (initial), $\alpha = 0.05$, $\tilde{\kappa}_m = 6$.
- **ResNet-32**:
  $\eta = 0.1$ (initial), $\alpha = 0.05$, $\tilde{\kappa}_m = 2$;
  $\eta = 0.3$ (initial), $\alpha = 0.05$, $\tilde{\kappa}_m = 24$.

