# OpenReview forum: "Accelerating SGD with momentum for over-parameterized learning"
_ICLR.cc/2020/Conference — Accept (Poster)_

### Official Review · AnonReviewer1 · 2019-10-12
**Official Blind Review #1**

**Rating:** 3

**Review:**

1) In the statement of theorem 1, what do you mean by “with probability one” and “convergence in expectation” together? The inequality (7) does not have any random variable anymore after taking the expectation. Need to explain more clearly this part.

2) Basically, all the results of this paper is based on the (or close to) strongly convex property. However, numerical experiments show for some non-convex functions, specifically for deep learning problems. It is unclear what kind of loss function the author(s) are using for training classification problems on MNIST and CIFAR-10. This could be softmax cross-entropy but not quadratic.

3) The theoretical results in this paper are not strong. The interpolation setting could make all solutions of the component function are the solution of the total loss function. In this situation, we know that stochastic algorithms could take advantage because of “automatic variance reduction”.

4) The result in this paper is quite incremental from the one in Vaswani et al 2019, “Fast and Faster Convergence of SGD for Over-Parameterized Models (and an Accelerated Perceptron)”. More discussion is needed if the author(s) think that it has significantly improvement.

5) It is true that Nesterov SGD is very efficient for training neural networks and MaSS may have some effect in practice. However, theoretical part needs to be improve. I would suggest to analyze for nonconvex problems or using the assumptions which are verifiable and reasonable for neural network.

The work may be potential, but in order to convince people to trust this algorithm, rigorous theory must be provided. Some experiments in the paper are not representing all scenarios that MaSS may not work.


**Experience Assessment:**

I have published in this field for several years.

**Review Assessment: Checking Correctness Of Derivations And Theory:**

I assessed the sensibility of the derivations and theory.

**Review Assessment: Checking Correctness Of Experiments:**

I assessed the sensibility of the experiments.

**Review Assessment: Thoroughness In Paper Reading:**

I read the paper at least twice and used my best judgement in assessing the paper.

---

> ### Author Response · Authors · 2019-11-07
> **Reply to Official Blind Review #1**
>
> Thank you for the comments.
>
> >>"The work may be potential, but in order to convince people to trust this algorithm, rigorous theory must be provided. Some experiments in the paper are not representing all scenarios that MaSS may not work.”
>
> We would like to point out a few key points:
> 1. we have rigorous theory to show the following two key results: non-acceleration of Nesterov SGD (Theorem 1) and the guaranteed acceleration of MaSS on quadratic and convex problems (Theorem 3 and 4).
> 2. From the practical point of view, we find that MaSS works well (better or at least as well as the standard methods) in all of our experiments, on both synthetic and real data, including  training of deep neural networks. Of course, it is not possible to explore all scenarios in a paper.
>
>
> We address your concerns one by one, as follows:
>
> >>”4) The result in this paper is quite incremental from the one in Vaswani et al 2019.”
>
> Comparison with the work (Vaswani et al 2019).
> Key differences:
> 1. The key theoretical result of non-acceleration of Nesterov SGD (Theorem 1) is new and is not known in the literature.
> 2. MaSS has provably guaranteed acceleration over SGD (Theorem 3 and 4). In contrast,  the convergence rate in (Vaswani et al 2019) can be significantly slower than that of SGD, even for quadratic problems with Gaussian distributed data. Moreover, in that setting MaSS matches the optimal rate of the original (non-SGD)  Nesterov’s algorithm. This is discussed in the third paragraph on page 3 and a detailed analysis is given in Appendix F.3.
> 3. We have analysis of the dependence of convergence rate of MaSS on batch size and the saturation phenomena for accelerated SGD. As far as we know, there is no such analysis in the current literature.
>
> >>"5) It is true that Nesterov SGD is very efficient for training neural networks and MaSS may have some effect in practice. However, theoretical part needs to be improve. I would suggest to analyze for nonconvex problems or using the assumptions which are verifiable and reasonable for neural network.”
>
> Even for plain SGD, there are very few results in the literature for non-convex problems (including neural networks). There are none, as far as we know, for accelerated/Nesterov SGD. We feel it is not reasonable to hold our paper to such a standard.
>
> >>"3) The theoretical results in this paper are not strong. The interpolation setting could make all solutions of the component function are the solution of the total loss function. In this situation, we know that stochastic algorithms could take advantage because of “automatic variance reduction”.”
>
> The interpolation setting has become a subject of significant interest recently as it appears that many deep models operate at (or close to) interpolation. See our discussion in the first paragraph of related work, as well as the cited papers (5;16;23;2) in our submission. Note that, as we show, even in the interpolation setting, Nesterov SGD does not accelerate over SGD.
>
> >>"1) In the statement of theorem 1, what do you mean by “with probability one” and “convergence in expectation” together?”
>
> Actually, we do not use the term “convergence in expectation” in Theorem 1 or elsewhere in the paper. We noticed the term “diverges in expectation” in Corollary 1, which will be replaced by “diverges” in the revision.
>
> >>"2) Basically, all the results of this paper is based on the (or close to) strongly convex property. However, numerical experiments show for some non-convex functions, specifically for deep learning problems. It is unclear what kind of loss function the author(s) are using for training classification problems on MNIST and CIFAR-10. This could be softmax cross-entropy but not quadratic.“
>
> We use softmax cross-entropy loss for the numerical experiments on neural networks, as is commonly used, and use quadratic loss for linear regression and kernel regression. Although our theoretical analysis is based on convex functions, our experiments show that MaSS practically works well for non-convex problems, such as training deep neural networks, including ResNet.

---

### Official Review · AnonReviewer2 · 2019-10-20
**Official Blind Review #2**

**Rating:** 8

**Review:**

This paper shows the non-acceleration of Nesterov SGD theoretically with a component decoupled model. Moreover, the authors introduce an additional compensation term and derive a novel optimization method, MaSS. MaSS is both theoretically and empirically proved to outperform Nesterov SGD as well as SGD.

Pros
1. It's amazing to see the great improvement introduced by the compensation term into the theoretical result of MaSS. Moreover, the authors generalize the setting of square loss function to other convex loss functions.
2. The encouraging result in Table 1 in EMPIRICAL EVALUATION shows the consistent outperformance of MaSS over SGD and Nesterov SGD regardless of the changing learning rates.

Cons
1. The discussion on why the zero eignvalue can be ignored in Section 4 is insufficient. "(stochastic) gradients are always perpendicular to W^*" seems not that obvious.
2. The empirical result merely involves two settings of learning rate: 0.01, 0.3. I suggest a wider range of learning rates to show the outperformance of MaSS.

Some typos
Last line of the first paragraph in INTRODUCTION: there is a redundant "can". 7th line of the 5th paragraph in INTRODUCTION: there is a reduntant "the" after "In this case".


**Experience Assessment:**

I have published one or two papers in this area.

**Review Assessment: Checking Correctness Of Derivations And Theory:**

I did not assess the derivations or theory.

**Review Assessment: Checking Correctness Of Experiments:**

I assessed the sensibility of the experiments.

**Review Assessment: Thoroughness In Paper Reading:**

I made a quick assessment of this paper.

---

> ### Author Response · Authors · 2019-11-08
> **Reply to Official Blind Review #2**
>
> Thanks for the encouraging comments.
>
> >>”1. The discussion on why the zero eignvalue can be ignored in Section 4 is insufficient. "(stochastic) gradients are always perpendicular to W^*" seems not that obvious.”
>
> We have added the discussion to the latest version of the paper. Please see section B.3 in the revision. We also corrected the typos.
>
>
> >>”2. The empirical result merely involves two settings of learning rate: 0.01, 0.3. I suggest a wider range of learning rates to show the outperformance of MaSS.”
>
> Thanks for your suggestion. We plan to conduct more experiments with different learning rates.

---

### Official Review · AnonReviewer3 · 2019-10-22
**Official Blind Review #3**

**Rating:** 8

**Review:**

The authors present a new first order optimization method that adds a corrective term to Nesterov SGD. They demonstrate that this adjustment is necessary and sufficient to benefit from the faster convergence of Nesterov gradient descent in the stochastic case. In the full-batch (non deterministic) setting, their algorithm boils down to the classical formulation of Nesterov GD. Their approach is justified by a well conducted theoretical analysis and some empirical work on toy datasets.

Positive points:
- The approach is elegant and thoroughly justified. The convergence to Nesterov GD when the batch size increase is comforting.
- The empirical evaluation, even if it is still preliminary and larger scale experiments will have to be conducted before the method could be widely adopted, are suitable and convincing.
- Some interesting observations regarding the convergence regimes (in respect to the batch size) are made. It would have been interesting to see how the results from fig3 generalize to the non convex problems considered in the paper.

Possible improvements:
- In H2, it is mentioned that the algorithm is restarted (the momentum is reset) when the learning rate is annealed. Was this also done for SGD+nesterov? Also, I think it is an important implementation detail that should be mentioned outside of the appendix
- Adam didn’t get the same hyper-parameter tuning as MaSS did. It is a bit disappointing, as I think the superior performance (in generalization) of non-adaptive methods would still hold and the experiment would have been more convincing. Rate of convergence is also not reported for Adam in fig 5.

I think this is definitely a good paper that should be accepted. I’m looking forward to see how it performs on non-toy models and if the community adopt it.


**Experience Assessment:**

I have read many papers in this area.

**Review Assessment: Checking Correctness Of Derivations And Theory:**

I assessed the sensibility of the derivations and theory.

**Review Assessment: Checking Correctness Of Experiments:**

I assessed the sensibility of the experiments.

**Review Assessment: Thoroughness In Paper Reading:**

I read the paper thoroughly.

---

> ### Author Response · Authors · 2019-11-08
> **Reply to Official Blind Review #3**
>
> Thank you for the encouraging comments.
>
> >>”In H2, it is mentioned that the algorithm is restarted (the momentum is reset) when the learning rate is annealed. Was this also done for SGD+nesterov? Also, I think it is an important implementation detail that should be mentioned outside of the appendix”
>
> The reported results in our submission is based on SGD+Nesterov without restart (as is the common practice). For comparison, we ran the same experiment (ResNet-32, lr=0.1) using SGD+Nesterov with restart, which gives slightly worse performance (91.65% accuracy, the average of 3 independent runs). We will clarify the setting of SGD+Nesterov in our revision.
>
> >>“Adam didn’t get the same hyper-parameter tuning as MaSS did. It is a bit disappointing, as I think the superior performance (in generalization) of non-adaptive methods would still hold and the experiment would have been more convincing. Rate of convergence is also not reported for Adam in fig 5.”
>
> We generated the training curves for Adam, please see the three plots at this link:
> https://anonymous.4open.science/r/37bcd717-2d8d-426f-a00c-9edd21e57647/
> These figures are in parallel to those in Fig.5. In the experiment we tuned the initial learning rate of Adam. The experimental setup of these figures is the same as that of fig.5 in the paper.

---

### Decision · Program_Chairs · 2019-12-19

**Decision:**

Accept (Poster)

**Comment:**

The authors provide an empirical and theoretical exploration of Nesterov momentum, particularly in the over-parametrized settings. Nesterov momentum has attracted great interest at various times in deep learning, but its properties and practical utility are not well understood. This paper makes an important step towards shedding some light on this approach for training models with a large number of parameters.